# Tracking aerosols and $SO_2$ clouds from the Raikoke eruption: 3D view from satellite observations

Nick Gorkavyi [1], Nickolay Krotkov [2], Can Li [3], Leslie Lait [1], Peter Colarco [2], Simon Carn[4], Matthew DeLand [1], Paul A. Newman [2], Mark Schoeberl[5], Ghassan Taha[2,6], Omar Torres[2], Alexander Vasilkov[1], Joanna Joiner [2]

[1] Science Systems and Applications, Lanham, MD, USA
[2] NASA, Goddard Space Flight Center, Greenbelt, MD, USA
[3] University of Maryland, College Park, MD, USA
[4] Michigan Technological University, Houghton, MI, USA
[5] Science and Technology Corporation, Columbia, MD, USA
[6] USRA, Greenbelt, MD, USA

**Correspondence:** Nick Gorkavyi (nick.gorkavyi@ssaihq.com)

**Abstract.** The June 21, 2019 eruption of the Raikoke volcano (Kuril Islands, Russia, 48°N, 153°E) produced significant amounts of volcanic aerosols (sulfate and ash) and sulfur dioxide ($SO_2$) gas that penetrated into the lower stratosphere. The dispersed $SO_2$ and sulfate aerosols in the stratosphere were still detectable by multiple satellite sensors for many months after the eruption. For this study of $SO_2$ and aerosol clouds we use data obtained from two of the Ozone Mapping and Profiler Suite sensors on the Suomi National Polar-orbiting Partnership satellite: total column $SO_2$ from the Nadir Mapper and aerosol extinction profiles from the Limb Profiler as well as other satellite data sets. We evaluated the limb viewing geometry effect (the "arch effect") in the retrieval of the LP standard aerosol extinction product at 674 nm. It was shown that the amount of $SO_2$ decreases with a characteristic period of 8-18 days and the peak of stratospheric aerosol optical depth recorded at a wavelength of 674 nm lags the initial peak of $SO_2$ mass by 1.5 months. Using satellite observations and a trajectory model, we examined the dynamics of an unusual atmospheric feature that was observed, a stratospheric coherent circular cloud of $SO_2$ and aerosol from July 18 to September 22, 2019.

## 1 Introduction

An eruption of the Raikoke volcano (see Fig. 1, 0.55 km summit altitude, uninhabited island at 48.29°N, 153.4°E) occurred on June 21, 2019 at 18:00 UT. The eruption was so strong that a cloud of ash and volcanic gases was ejected to a height of 17-19 km (Gorkavyi et al., 2020), putting a significant part of the Raikoke volcanic plume into the stratosphere and above the heights of meteorological water and ice clouds and the tropopause, which is ~ 10-12 km for such northern latitudes (see Fig. 1). The Microwave Limb Sounder (MLS) satellite data for June 23-27, 2019 show that the observed parts of the dispersed $SO_2$ cloud had heights between 11 and 18 km, with a peak concentration at 14 km. The maximum volcanic cloud top height increased by more than 6 km within 4 days after the eruption due to aerosol–radiation interaction (Muser et al., 2020).

The study of dispersed volcanic clouds and the transformation of their $SO_2$ into sulfuric acid aerosols is interesting for several reasons. Firstly, clouds of ash and sulfate aerosol are aviation hazards (Carn et al., 2009; Krotkov et al., 2014). Secondly, the lifetime of volcanic $SO_2$ depends on its injection height, $SO_2$ uptake on ash, and the concentrations of oxidants that vary with season and location (Guo et al., 2004; Krotkov et al., 2010; Beirle et al., 2014; Zhu et al., 2020). $SO_2$ from volcanic eruptions has a longer lifetime if it is injected into the stratosphere (Carn et al., 2016). $SO_2$ in the stratosphere is converted to sulfate aerosol, which interacts with the long-lived Junge layer—the naturally occurring background

stratospheric aerosol layer. Furthermore, the dynamics of the Junge layer itself is not well understood. In addition, stratospheric aerosols are an important climate forcing factor, because aerosols modify both the shortwave and longwave radiation in the atmosphere and reaching the Earth's surface (Toohey et al., 2019; von Savigny et al., 2020). Thus, each case of volcanic injection of large amounts of $SO_2$ into the stratosphere is of great interest to climate scientists (Robock 2000;
Foster et al., 2008). Our ability to study volcanic $SO_2$ has evolved in recent decades along with satellite remote sensing technology (Krueger 1983; Krueger et al., 2000, 2008; Bovensmann et al., 1999; Guo et al., 2004; Carn et al., 2003, 2007, 2008, 2009, 2017; Clerbaux et al., 2009; Clarisse et al., 2010, 2013; Penning de Vries et al., 2014; Sandvik et al., 2019; Hedelt et al., 2019; Theys et al., 2019; Fisher et al., 2019). The spectral data obtained from satellites has made it possible to analyze in detail transient volcanic $SO_2$ clouds, such as of the eruptions of Pinatubo (Bluth et al., 1992; Guo et al., 2004;
Fisher et al., 2019), El Chichon (Krueger 1983; Krueger et al., 2008), and Kasatochi (Bourassa et al., 2010; Prata et al., 2010; Krotkov et al., 2010; Clarisse et al., 2011).

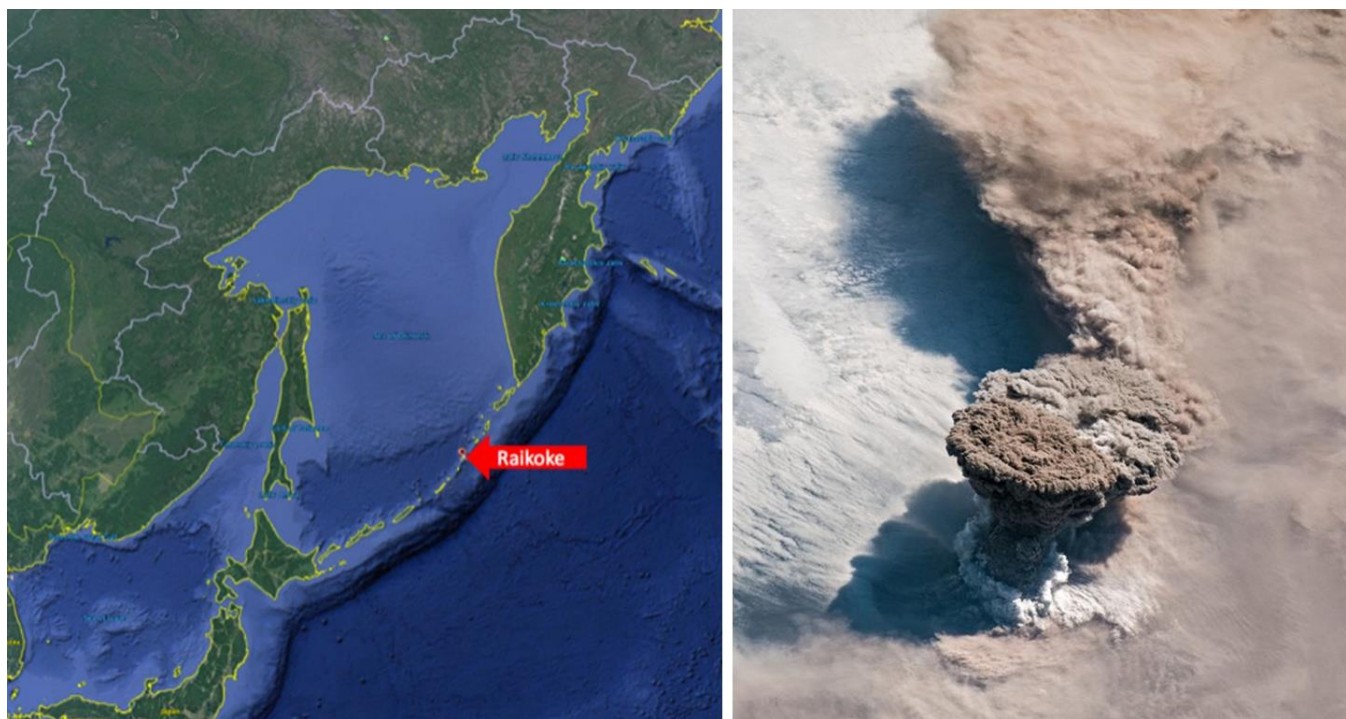

**Figure 1: Left: Raikoke volcano (Kuril Islands, Russia, 48ᵒN, 153ᵒE) (Google Maps, Imagery: NASA). Right: International Space Station (ISS) photo of the Raikoke eruption (06/21/2019, 22:45 UTC), which shows that the volcanic plume rose much higher than the layer of tropospheric clouds (Photo: iss059e119250 June 22, 2019, ISS/NASA, https://visibleearth.nasa.gov/images/145226/raikoke-erupts/145230w)**

The 2019 eruption of Raikoke was well observed by a number of satellite sensors including the Ozone Monitoring Instrument (OMI), the Tropospheric Monitoring Instrument (TROPOMI), the Ozone Mapping and Profiler Suite (OMPS) Nadir Mapper (NM) and Limb Profiler (LP) and the Cloud-Aerosol Lidar with Orthogonal Polarization (CALIOP). Studying the nadir passive data of OMPS NM and TROPOMI, our attention was drawn to an interesting phenomenon, the presence of compact long-lived stratospheric $SO_2$ clouds, hereafter referred to as coherent circular clouds or CCC, each about 300 km in
diameter. The first was noted on June 29, 2019 centered at 51°N, 157°W (near Alaska), and the second was observed starting on July 18, 2019 near Kamchatka. The first of these moved towards the North Pole while the second moved south to ~ 20-30°N, where it remained for more than two months, having made almost three complete revolutions around the Earth.

A dense cloud of $SO_2$ and aerosol that formed after the eruption of the Raikoke volcano spread over a week at latitudes
above 40°N (see TROPOMI $SO_2$ map on June 29, 2019 in Fig. 2). Atmospheric currents stretched the cloud into long jets,
twisted them with spirals and even formed CCCs, as shown in Fig. 2 in the region near the coordinates 52°N, 156°W. To
analyze in detail the evolution and dynamics of the Raikoke volcanic cloud, we examine 1) the amount and height of $SO_2$
and aerosol emitted during the eruption using satellite nadir and limb data; 2) the mutual evolution of $SO_2$ and aerosol for
100 days, starting from the moment of the eruption with limb data using an adjustment correcting for limb viewing geometry
effects; 3) the propagation of the CCCs with a case study in July 2019 using a trajectory model.

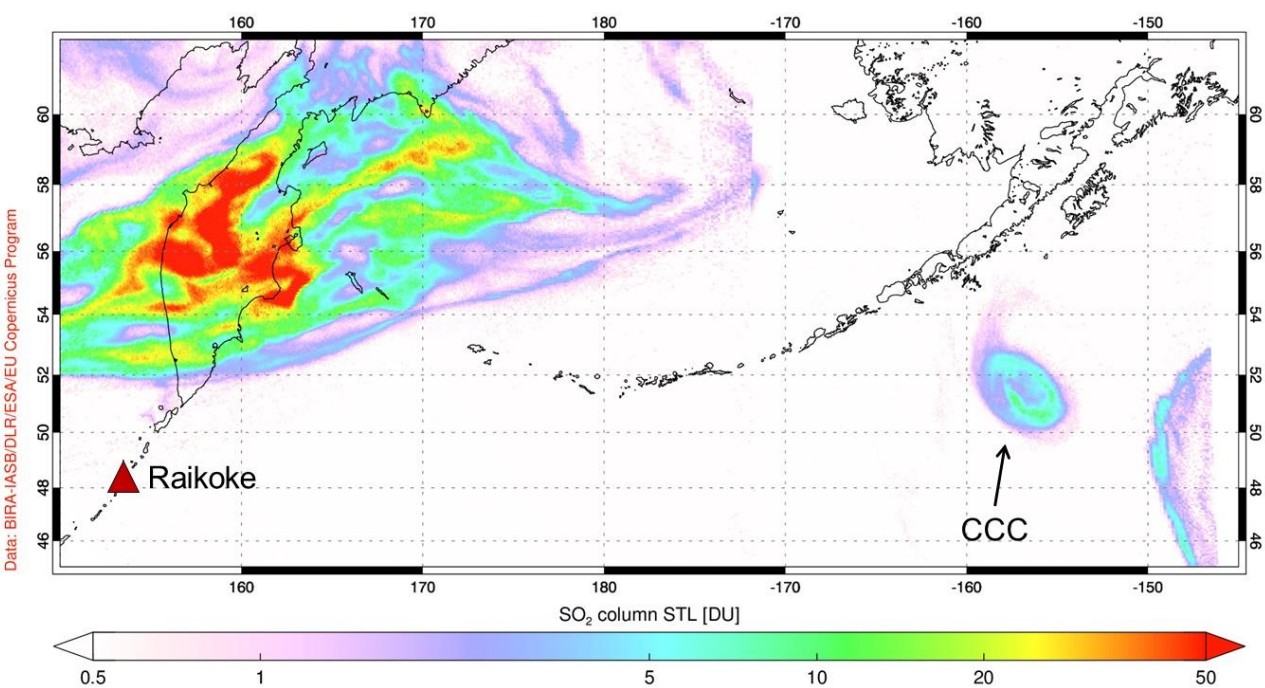

**Figure 2: TROPOMI image of total column $SO_2$ (log scale), a week after the Raikoke eruption (Image: courtesy
Copernicus TROPOMI/Sentinel-5Precursor $SO_2$ data, https://so2.gsfc.nasa.gov/). The triangle represents the location
of the Raikoke volcano. A coherent circular cloud is seen on June 29, 2019 centered at 51°N, 157°W. Note that this
plot crosses the date line and so includes data from June 29 (where the CCC is located) and June 30 (UTC).**

**2 Data and Methods**

In this section, we detail the instruments and methods used to 1) examine $SO_2$ and aerosol distributions obtained from
passive nadir sensors for $SO_2$ and limb and lidar for aerosol; 2) calculate zonal mean values of $SO_2$ and aerosol as a function
of a height using nadir data for $SO_2$ and limb data for aerosol; 3) employ trajectory modeling for analysis of individual
clouds.


## 2.1 Satellite Mapping of SO$_2$

We use SO$_2$ total column data obtained from two satellite spectrometers: 1) TROPOMI (see Fig. 2, Theys et al., 2017, 2019)
on the Sentinel 5 precursor satellite and 2) OMPS Nadir Mapper (NM) on the Suomi National Polar-orbiting Partnership (SNPP) satellite (Li et al., 2013, 2015; Zhang et al., 2017). Both instruments observe the Earth's backscattered radiance and solar irradiance at hyperspectral ultraviolet (UV) wavelengths with two-dimensional charge-coupled devices (CCDs), measuring in the spectral domain in one dimension and in the spatial domain (across a satellite track) in the other dimension. The satellite motion provides measurements along the satellite track. The spatial resolution of TROPOMI (3.5 km by 7 km)
is much finer than that from OMPS NM measurements made in the nominal mode (50 km by 50 km) and the spectral resolution is higher for TROPOMI as well (~ 0.5 nm as compared with ~ 1 nm for OMPS NM). The satellite swaths are wide (2700 km for TROPOMI and 2800 km for OMPS NM), providing nearly global daily coverage. Column SO$_2$ in Dobson Units (1 DU= $2.69*10^{16}$ molecules cm$^{-2}$) is retrieved from hyperspectral solar backscatter at UV wavelengths (312-390 nm). TROPOMI uses a spectral fitting algorithm based on differential optical absorption spectroscopy (DOAS) (Theys et al.,
2017, 2019). OMPS uses spectral fitting approach with a principal component analysis (PCA) scheme (Li et al., 2013, 2017). For large volcanic SO$_2$ signals like Raikoke, comparisons between TROPOMI and SNPP/OMPS so far (for several eruptions) show little bias, with the total SO$_2$ mass estimates from the two normally agreeing to within 5-10% (with the exception of the very early stages of large eruptions, where the density of SO2 and/or volcanic ash is too high to be fully accounted for in operational algorithms). For retrieval noise on a pixel-to-pixel basis, SNPP/OMPS SO$_2$ (for stratospheric
clouds) is less than 0.1 DU. TROPOMI's noise on a pixel-by-pixel base is several times greater, but once TROPOMI pixels are averaged to OMPS footprints, the noise is reduced by ~30%. We also used Ozone Monitoring Instrument (OMI) on NASA's Aura satellite as an additional data source.

## 2.2 Aerosol profile data

We use backscatter data from CALIOP (Fairlie et al., 2014), OMPS Limb Profiler (LP) (Loughman et al., 2018; Chen et al., 2018) and Stratospheric Aerosol and Gas Experiment instrument onboard the International Space Station (SAGE III/ISS). The OMPS LP V1.5 aerosol retrieval algorithm is described Sects. 2 and 3 of Chen et al. (2018). The height of an aerosol cloud can be estimated with both OMPS LP and CALIOP. We use the CALIOP lidar images for aerosol, specifically 532 nm
total attenuated backscatter signal with a spatial resolution of 40 km along track and a vertical resolution of 120 m (altitude < 20 km) and 360 m (altitude > 20 km).

OMPS LP hyperspectral measurements capture aerosol data with a sampling of ~1 km in altitude. The actual cross-track FOV of each OMPS LP slit is ~4 km (Dittman et al., 2002). OMPS LP views the atmosphere in a backward direction along
the orbit track with three vertical slits, one (central) aligned with the orbit track and the other two (left and right) separated by ±4.25° horizontally. Here, we use aerosol data from version 1.5 (Chen et al., 2018).

Limb measurements of the atmosphere that view an altitude $H$ at the tangent point also view altitudes above $H$ in the foreground and background of the line of sight. However, the converse situation is also true: an object at a fixed altitude $H$
will appear to be at a lower altitude $h$ if it is located closer or farther than the tangent point. This is shown schematically in Figure 3a, where a single cloud is located at true altitude $H$ and 5 successive LP measurement events (each event generates a new profile) are illustrated. For event C, the cloud location is at the tangent point, and the apparent altitude $h$ equals the true altitude $H$. For events A and B, the cloud position is closer to the LP instrument than the tangent point, and $h$ is less than $H$. For events D and E, the cloud position is farther from the LP instrument than the tangent point, and again $h < H$.

Plotting successive altitude profiles of LP (e.g., extinction coefficient) as function of tangential latitude when a vertically limited feature (such as a cloud) is present gives the impression of an "arch" in the data. The arch effect is observed when the length of the visible part of the cloud is less than ~1100 km (at a cloud height of 25 km). Figure 3b shows cloud $F_0G_0$, 1 km thick and 226 km long, centered above tangential point T. Due to the curvature of the globe, such a cloud has an
observed thickness of 2 km (see Figure 3b). If we take a cloud 226 km long and with a real thickness of N km, then the

observed cloud thickness will be N+1 km. Thus, the real average height of a thin (1-2 km) cloud is underestimated by 0.5-1 km even under the most optimal observation conditions. Consider a cloud FG, the center of which is displaced from the tangential point by 273 km (or by $\phi\sim2.5$ degrees). The real height of the FG cloud is 24-25 km, but its observed height varies from 13 to 22 km. If we consider the $F_0G$ cloud with a length of 499 km, then its real height above the earth's surface will be

24-25 km, and the observed height is 13-25 km. Let us take into account that the limb profiler assigns the latitude of the tangential point to any extended cloud. Therefore, a single cloud shown in Figure 3a in five different observed positions, instead of one real geographic latitude, receives several "observed" latitudes, which creates an arch effect. Let the region $F_0G_0$ be a gap in a continuous cloud. Then this gap, together with the arch effect, will lead to a decrease in the maximum observed height of the cloud layer by 1 km (see Figure 3b).


Figure 4, showing an OMPS LP extinction profile track across the volcanic plume, illustrates the arch effect. The maximum values (yellow) appear to vary in altitude by 2-3 km due to the projection effect shown in Fig. 3. Note that this effect will be present for high, thin clouds such as cirrus clouds as well as for aerosol plumes. If we believe that these lower altitude values do not represent a true aerosol signal, we need to apply a correction in order to accurately determine overall aerosol

loading.

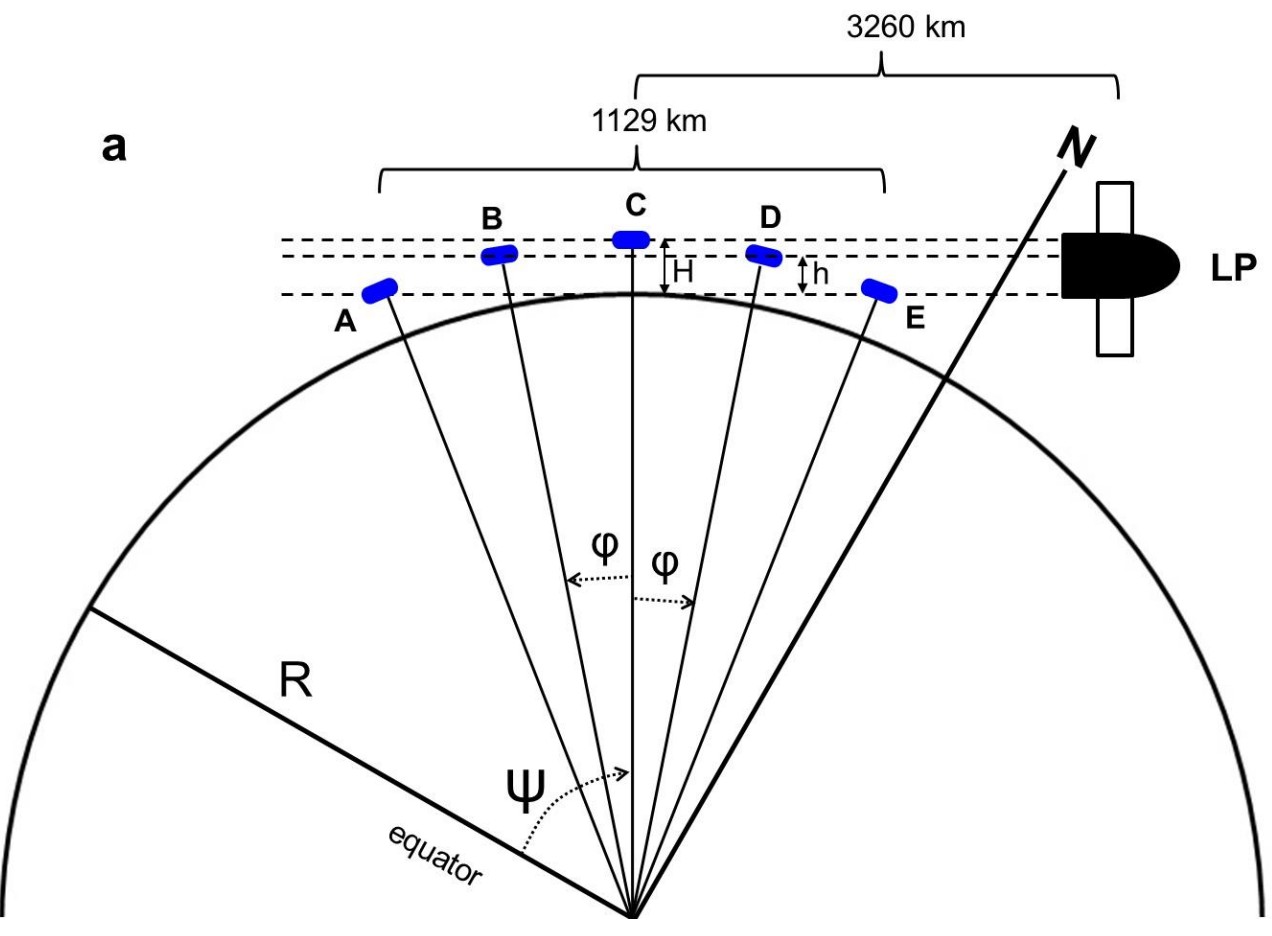


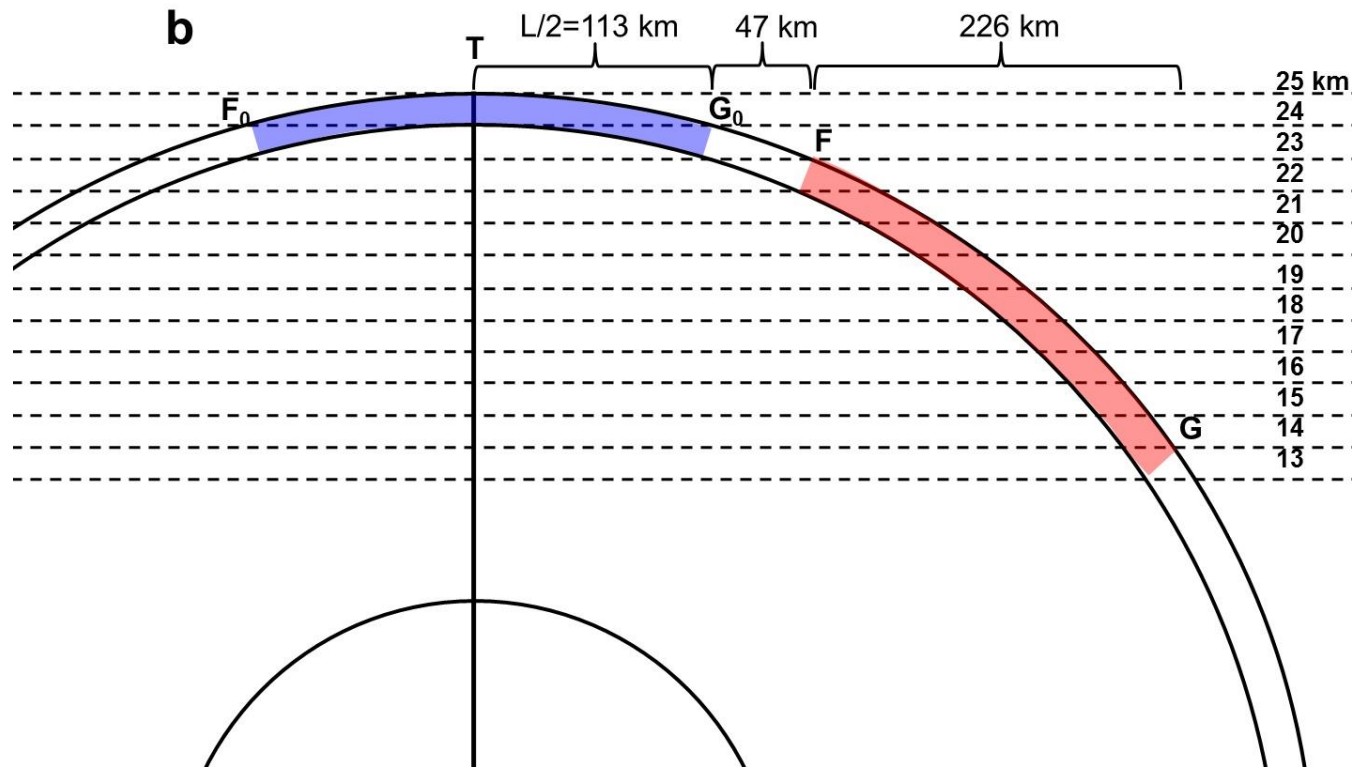

**Figure 3: (a) The diagram for observations of a limb sensor, showing the relationship between the observed (h) and real (H) heights of one cloud in five positions, as well as φ – the angular displacement of the position of the cloud from the tangential point T. Ψ is a latitude. The figure is not to scale. (b) Ratio of real and observed heights for a cloud 1 km thick and 226 km long, the center of which is located at different distances from the tangential point.**

The relationship between true cloud height $H$ and apparent cloud height $h$ in Figure 3 is given by a simple function (DeLand and Gorkavyi, 2021):

$$\cos \varphi \; = \frac{R+h}{R+H},\qquad\qquad (1)$$

where $R$ is the radius of the Earth and $\varphi$ is the angular displacement. Table 1 gives some specific examples of the angular displacement predicted for specific combinations of $H$ and $h$. This displacement is approximately equal to a shift in latitude for a sun-synchronous satellite such as S-NPP (inclination = 98.9°) outside the polar regions. From this table, we see that for a true cloud at $H = 25$ km, the apparent cloud signal at h = 15 km would be displaced by ~3.2° both earlier and later along the orbit. This total separation of ~6.4° is consistent with the "arch width" at 15 km shown in Fig. 4. Note also that because of differences in overall path length, we expect events A and B in Fig. 3 (left branch of "arch" in Fig. 4) to have a stronger signal than the corresponding events D and E at the same apparent altitude. We can therefore use Eq. (1) to calculate and apply a correction for determining the magnitude and position of an aerosol cloud (we believe that all parts of the arch below the real height are artifacts, so the value of the extinction coefficient for them should be equal to zero).

Table 1. The observed height $h$ and angular displacement $\varphi$ for the cloud with true height $H$.

| $h\backslash H$ | $H = 10$ km | 15 km | 20 km | 25 km | 30 km |
|---|---|---|---|---|---|
| $h = 0$ km | $\varphi = 3.21°$ | 3.93° | 4.53° | 5.07° | 5.55° |
| 5 km | 2.27° | 3.21° | 3.93° | 4.53° | 5.07° |
| 10 km | - | 2.27° | 3.21° | 3.92° | 4.53° |
| 15 km | - | - | 2.27° | 3.20° | 3.92° |
| 20 km | - | - | - | 2.27° | 3.20° |
| 25 km | - | - | - | - | 2.26° |

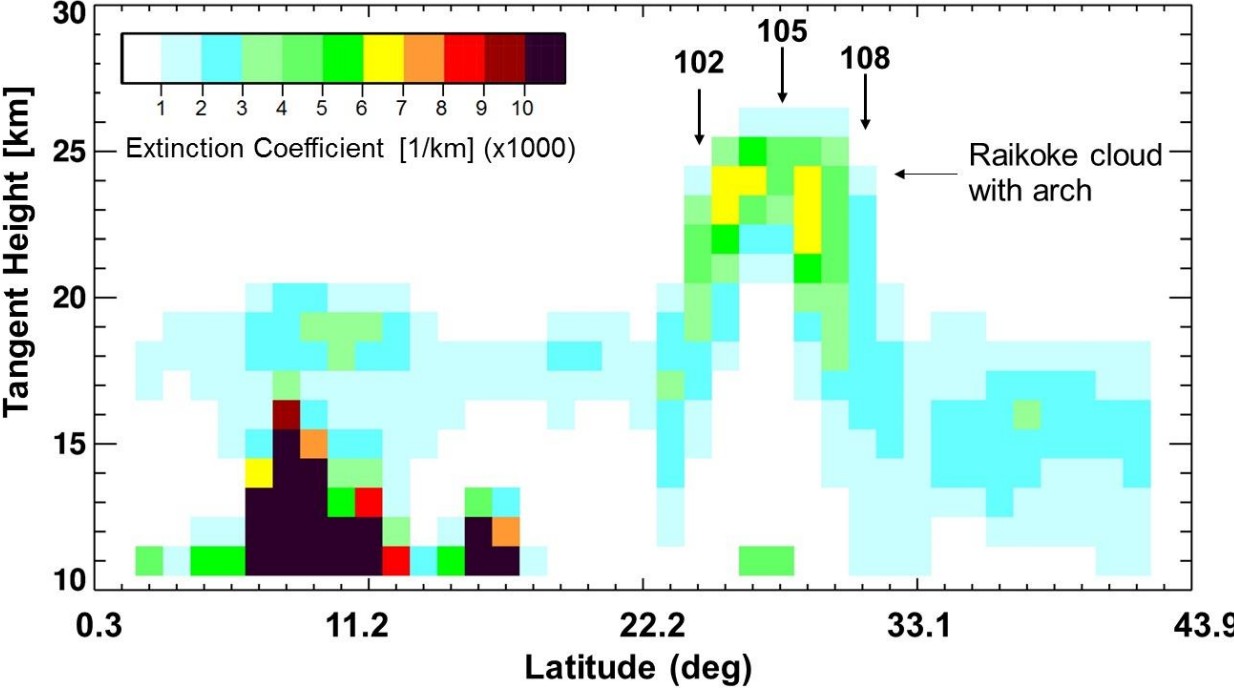


**Figure 4a: The extinction coefficient profiles (674 nm) from OMPS LP for the orbit 40636 (central slit, same orbit as in Fig. 10, which shows the profiles for events 102, 105, 108) for August 31, 2019. To accurately calculate the extinction coefficient it is necessary to take into account the arch effect, which is clearly visible (OMPS LP data).**

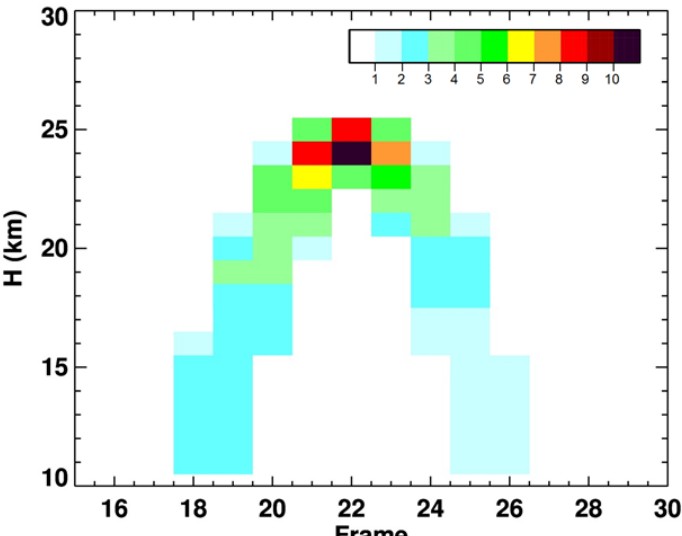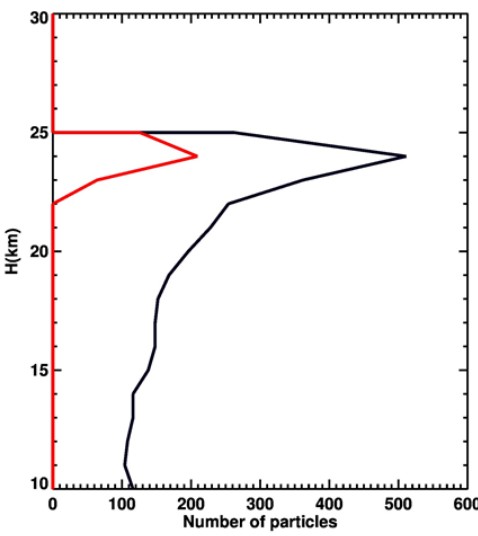

**Fig 4b. (Left) An arch that appears during the limb observation of an aerosol cloud 2 km thick (H = 23-25 km) and 2°**
**long. Each frame corresponds to a satellite orbital shift by 1°. Radiance from aerosol is proportional to the number of**
**observed particles, taking into account the distance to them (the left part of the arch is slightly brighter than the**
**right, because on frames 17-21 the cloud was closer to the satellite than on frames 23-27). The radiance units are**
**arbitrary. (Right). The red line is the profile of a cloud of 400 particles, which is observed on frame 22 (middle of the**
**arch in Fig. 4a). The black line is the number of visible particles, summed over all frames, that is, over the entire**
**arch.**



The arch in Fig. 4a is formed by several measurements of one CCC, which has a true height of 23-25 km (see the analysis of this CCC in Sect. 3.2). Parts of the arch (between events 102 and 108, indicated with arrows in Figure 4a) below 23 km are artifacts of multiple registrations of the same object and should be removed when calculating the total amount of aerosol at given latitudes or heights. The low stratospheric aerosol layer at an altitude of 18-20 km decreasing with latitude to 15-16 km (blue color) is the background layer. It merges in frames 101-109 with an apparent volcanic cloud arch. Note that the CCC has been the southernmost part of the Raikoke plume since late July 2019 (see section 3.3). The tropical cloud at latitude ~ 8°N extends up to 16 km altitude.


The reality of the discussed "arch effect" is confirmed by simple modeling. Arch model in Fig. 4b was obtained by direct modeling of a cloud of 400 particles located at the nodes of a uniform grid (20 particles are distributed at a length of 2 degrees, and 20 rows of particles are uniformly distributed along the radius in the range of 23-25 km). The model does not use radiative transfer models. Fig. 4b (left) only the distance between the particles and the satellite is taken into account, while Fig. 4b (right) shows the simply visible (for the limb sensor) number of particles at a given altitude h (in 1 km step). The red line corresponds to a one-time observation of a cloud located at the tangent point at an altitude of h = 22-25 km (an increase in the apparent thickness of the cloud by 1 km is associated with the curvature of the Earth, which is why the cloud itself turns out to be curved - see Fig. 3). That is, it is the most realistic observation of the cloud at its optimal location. The black line shows the sum of the cloud particles observed at different times. As a result of this summation, the number of particles visible at a given observed height h (which differs from the real constant cloud height H = 23-25 km) turns out to be overestimated. Therefore, when analyzing the picture in Fig. 4, we must remember that it is composed of frames received at different times, so the one cloud will be registered many times. The same effect of multiple registrations will be observed for clouds of any complexity and configuration, including a uniform aerosol layer, because any cloud can be divided into a large number of elementary pieces, similar to a simple compact cloud considered in Fig. 4b. Obviously, the arch effect for a




spherically uniform aerosol layer should be fully compensated by the 1D retrieval model. But the further the real system is
from the spherical symmetry, the more difficult it will be to take into account the "arch effect".

The considered model of the "arch effect" does not depend on the specific model of radiation transfer and the methods of retrieval of the spatial distribution of aerosol. Therefore, for specific limb sensors (OMPS-LP, SAGE III, OSIRIS), it is necessary to evaluate how accurately the available retrieval packages handle compact clouds. This is especially true for 1D
retrieval methods, which assume spherical symmetry of the atmosphere and which are used to obtain aerosol extinction in the OMPS/LP. To estimate the possible retrieval uncertainty due to the "arch effect", we apply a simple compensation method for one specific case of the Raikoke aerosol cloud (see next section). This compensation method assumes that the considered aerosol clouds form compact clusters or a highly heterogeneous aerosol layer, and that the arch effect was not taken into account in retrieval. Thus, this example should be regarded as a maximal estimation for the "arch effect". Where
this assumption is not valid, our correction will be overestimated, as, for example, happened with the correction of the background aerosol value (see Fig. 5), which was observed before the Raikoke eruption.

The principle of our posterior algorithm for estimating the "arch effect" is as follows: the data in the uppermost pixel of the arch are considered true. For a given pixel, artifact extinctions are calculated for pixels in the side branches of the arch.
These artifact extinctions are subtracted from the initial arch extinction. The procedure is repeated for the second highest pixel in the already modified arch image. This continues until the side branches of the arch completely disappear. The top pixels with the corrected extinction are summed into the cloud with the corrected data. We consider this procedure of correction only as an estimate, which shows the possible significance of the arch effect.

The arch effect is a specific example of the effects of inhomogeneity along the line of sight that is an issue for all types of limb sounders. One way to account for such effects is to use a two-dimensional (2D) radiative transfer model (RTM) that is able to account for such effects along with multiple observations in a tomographic retrieval (e.g., Livesey et al., 2006; Zawada et al., 2018; Loughman et al., 2018). Instead, we have developed an a posteriori adjustment method that is effective in correcting for the arch-type effects that appear with isolated features within the lines of sight for a series of observations.

The arch effect characteristic of the limb observations should be taken into account when calculating the optical thickness of aerosol clouds, determining a characteristic cloud height, or using calibration heights of ~ 45 km. Calibration heights should be free of aerosol and they are indeed free of aerosol but can be contaminated with polar stratospheric and mesospheric clouds (PMCs) and other stratospheric aerosols. For example, PMCs located at an altitude of 80-85 km, due to the arch
effect, can be projected onto the calibration heights. Such contamination of the calibration heights distorts the whole picture of aerosol distribution with height.
SAGE III/ISS makes direct measurements of aerosol extinction through the attenuation of the solar beam during sunrise and sunset.  The vertical resolution of these measurements is about ½ km.  Because they are occultation measurements, SAGE observations are less frequent than OMPS-LP.

Using the color ratio (extinction at 512 nm to extinction at 1022 nm) along with the extinction 1022 nm, clouds can be removed from the SAGE observations to produce average aerosol extinctions (Thomason and Vernier, 2013).

**2.3 The NASA ftraj trajectory model**

The "ftraj" trajectory model from NASA's Goddard Space Flight Center Atmospheric Chemistry and Dynamics Laboratory uses a fourth-order Runge-Kutta integration scheme to track parcels isentropically, with optional diabatic adjustments (Schoeberl and Sparling, 1995). The model is driven with winds at 1/4° horizontal resolution and spaced every six hours from the Goddard Earth Observing System (GEOS) forward processing system produced by the NASA Global Modeling and
Assimilation Office (GMAO). Each model run was initialized with 3000 parcels distributed at random horizontal and vertical positions within in a cylinder of radius 150 km centered about the location of the observed cloud and stretching between the lowest and highest potential temperature ($\theta$) values of the observed cloud. Those $\theta$ values were derived by

interpolating the GEOS fields to the parcels' initial longitude, latitude, altitude, and time. Both the starting and ending cloud observations extended through a range of altitudes and hence $\theta$ values.

**3 Results**

The sensitivity of the satellite data (OMPS LP, CALIOP) we examined is such that the aerosol cloud from the Raikoke volcano was observable for many months following the eruption. Thus, a synergistic study of Raikoke volcanic emissions using various satellite-based instruments provides a good opportunity to study the dynamics of the volcanic cloud long-term dispersion. For quantitative analysis, we will use 2D $SO_2$ vertical column density (VCD) maps obtained by OMPS NM, as
well as vertical profiles of aerosol extinction that are obtained from the OMPS LP.

**3.1 $SO_2$ and aerosol evolution**

The retrieved Raikoke $SO_2$ mass increased for 2 days after the eruption and then exponentially decayed with an e-folding
time scale of 8-18 days. The amount of stratospheric $SO_2$ decreases due to photochemical conversion to sulfuric acid though gas - phase reaction with the hydroxyl radical, OH. Sulfuric acid nucleates new particles and condenses onto pre-existing particles to form long-lived stratospheric sulfate aerosol. Figure 5 shows the change in the total $SO_2$ mass (kt) retrieved by the OMPS NM and the altitude averaged (13-18 km) aerosol extinction coefficient retrieved with the OMPS LP in the latitude range 45°-65°N after the Raikoke eruption. The aerosol extinction data in Fig. 5 are shown before and after
removing the "arch effect". These estimates show the potential significance of the arch effect. As we discussed earlier, the arch effect estimation assumes high cloud heterogeneity and should be considered the maximum estimation, as indicated by the arrow bars on Fig.5. For more accurate calculations, the arch effect should be investigated using a 2D RTM.

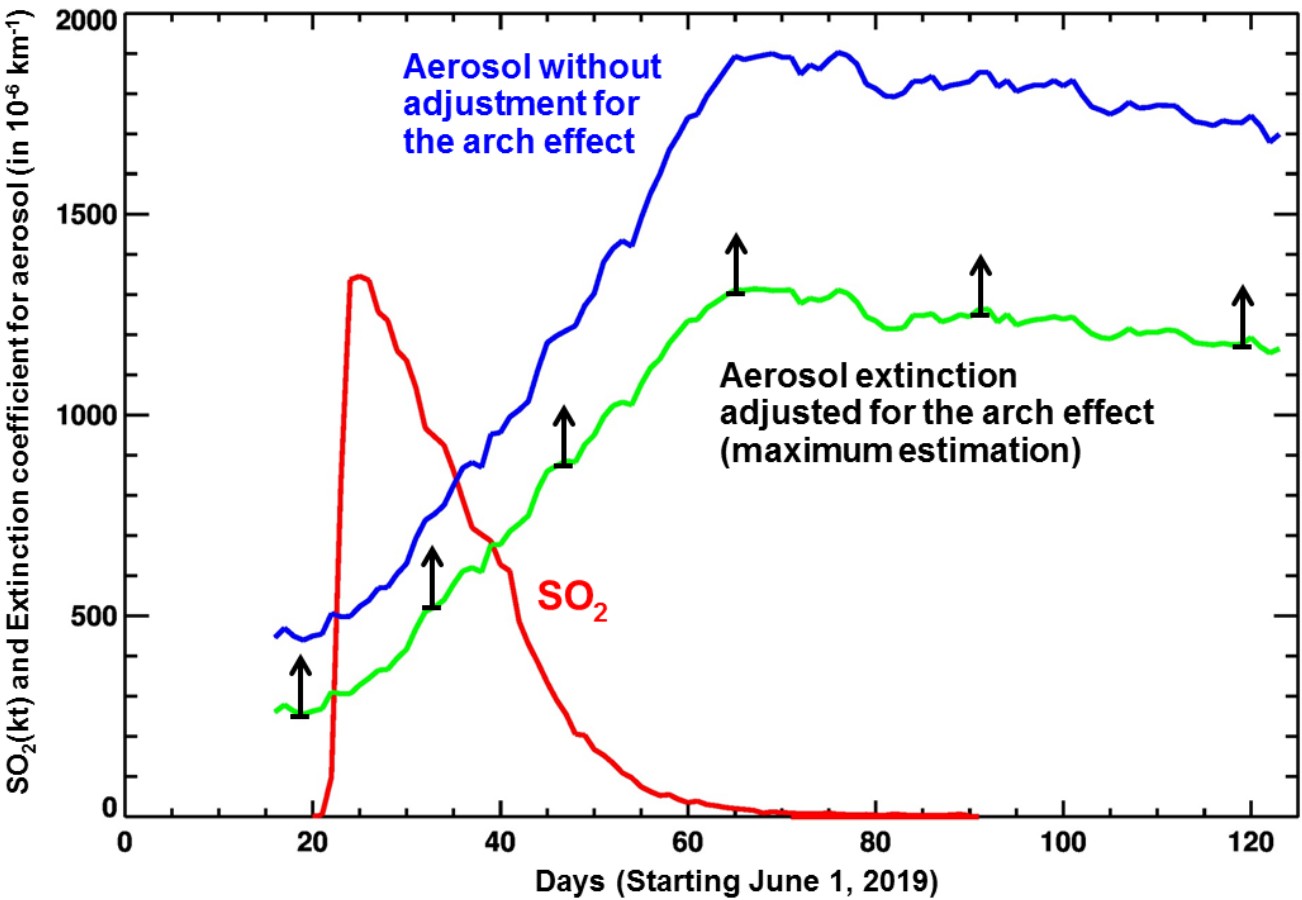

**Figure 5: The daily zonal mean (45-85°N) SO$_2$ mass (assuming a cloud height of 13 km) and the average aerosol extinction coefficient at 675 nm (summed up over 13-18 km and divided by 6 (km) to get the average extinction coefficient). The arrow bar shows the maximum possible error in calculating the aerosol extinction due to the arch effect.**

Figure 5 shows that retrieved SO$_2$ mass increases for 2 days after the eruption during rapid dispersion of the fresh opaque volcanic cloud. Following this initial dispersion, the SO$_2$ mass decreases exponentially due to chemical conversion to sulfate.
Aerosol extinction has opposite behavior; it increases by approximately a factor of 4 as the concentration of SO$_2$ decreases, reaching a maximum ~50 days after the eruption. After that, the aerosol extinction starts to decrease due to gravitational sedimentation and other processes, but very slowly (from OMPS LP and NM data). Adjustments for the arch effect do not change the overall temporal pattern.

As can be seen from Fig. 6, the total SO$_2$ mass decreases exponentially with a variable (8-18 day) timescale. Krotkov et al.
(2010) derived a volcanic SO$_2$ mass decay of ~ 9 days following the August 8, 2008, Kasatochi eruption (30°N–90°N) using the Aura OMI. The overall apparent e-folding time for Raikoke is ~ 8-10 days (Fig. 6, solid black lines), in good agreement with the Kasatochi e-folding time. However, we see two different regimes in Raikoke SO$_2$ clouds with significantly different estimated e-folding time, as discussed below.

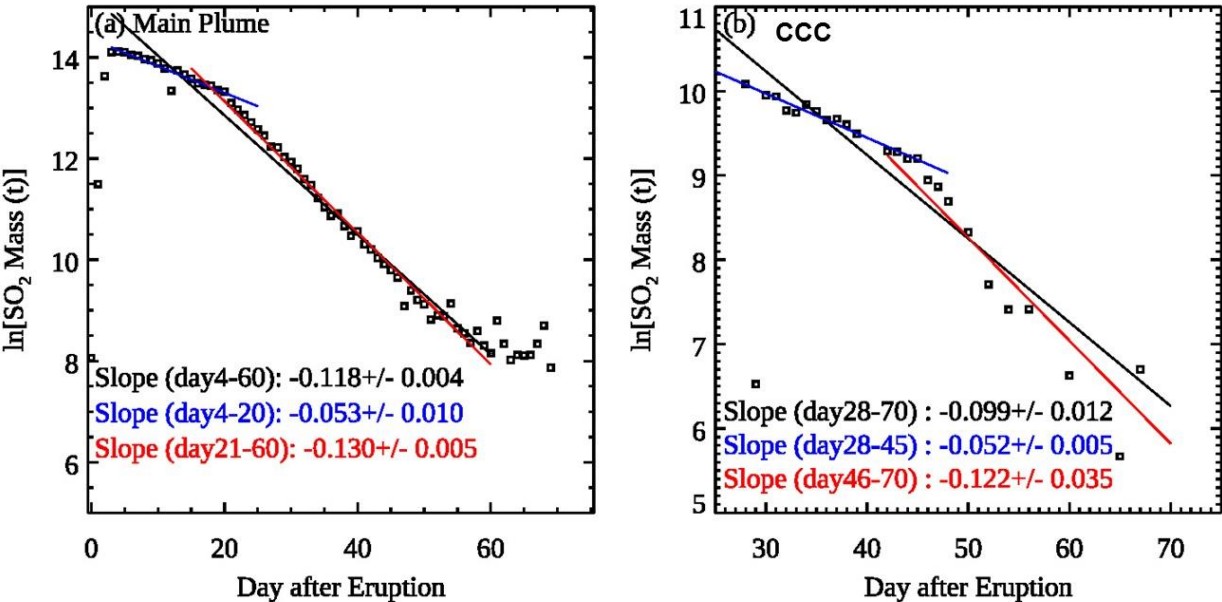

**Figure 6: Time series of the logarithm of the estimated total SO$_2$ mass (in tons) within (a) the main plume (45-85°N) and (b) the CCC.**

The left panel in Fig. 6 shows that during the first 20 days after the eruption, the SO$_2$ mass decreases with a longer apparent e-folding lifetime of ~19 days, compared to the later period (days 21 to 60), when the decrease in SO$_2$ mass accelerates with the apparent e-folding lifetime ~8 days. The initial estimate of the e-folding lifetime of ~19 days, derived from the earlier
stage after the eruption, probably represents the true time scale for chemical conversion of SO$_2$ into sulfate aerosol on altitudes >20 km (Carn et al., 2016). Following the drop of the SO$_2$ column amounts, the SO$_2$ detection limit of the OMPS instrument becomes a more important factor, leading to faster apparent decay of the total observed mass of SO$_2$. As the plume spreads out to larger areas, and more and more pixels with SO$_2$ fall below the detection limit of the OMPS NM sensor (OMPS in the stratosphere can typically detect 0.2-0.3 DU of SO$_2$), the apparent decay rate becomes larger, likely reflecting
the combined effects of the chemical SO$_2$ loss and the diminished OMPS sensitivity to weaker SO$_2$ signals. Figure 6b shows a similar pattern for change in SO$_2$ mass within the CCC. For the period between 28 and 45 days after the eruption, the e-folding lifetime is ~19 days. This suggests that the time after the eruption does not play a big role here, only the compactness of the CCC with a high level of SO$_2$ is essential. As in case Fig. 6a, the second phase of the evolution of the CCC is characterized by a faster apparent (the e-folding lifetime ~ 8 days) rate of decrease in the recorded SO$_2$.

**3.2 Stratospheric aerosol from limb observations**

Figure 7 shows the time-latitude evolution of the zonal average aerosol extinction at different heights (without adjustment for the arch effect). Most of the aerosol is transported poleward at 14.5 km (Fig. 7a), and the effects of the eruption lasted for almost a year. In addition, Figures 7b and c also show the aerosol transport to subtropics and tropics at higher altitudes. Increased aerosol loading in the lower stratosphere can also be attributed to two pyroCumulonimbus (pyroCb) events that
took place before and after the eruption, Alberta fires (June 18) and Siberian fires (July 2) (Kloss et al., 2021). OMPS LP detected both plumes in the stratosphere at 12-13 km, although it became difficult to separate them from Raikoke plume once it spread around the NH. The zonal mean aerosol extinction profile between 20˚N and 90˚N shows the vertical transport of the Raikoke plume to higher altitudes and its persistence in the lower stratosphere (Fig. 8). The top panel in Fig. 8 shows that the maximum altitudes of the plume are around 25 km, when the plume penetrates the tropics. The plume altitude is
derived using the OMPS LP cloud algorithm, which can identify enhanced aerosol layers in the stratosphere.

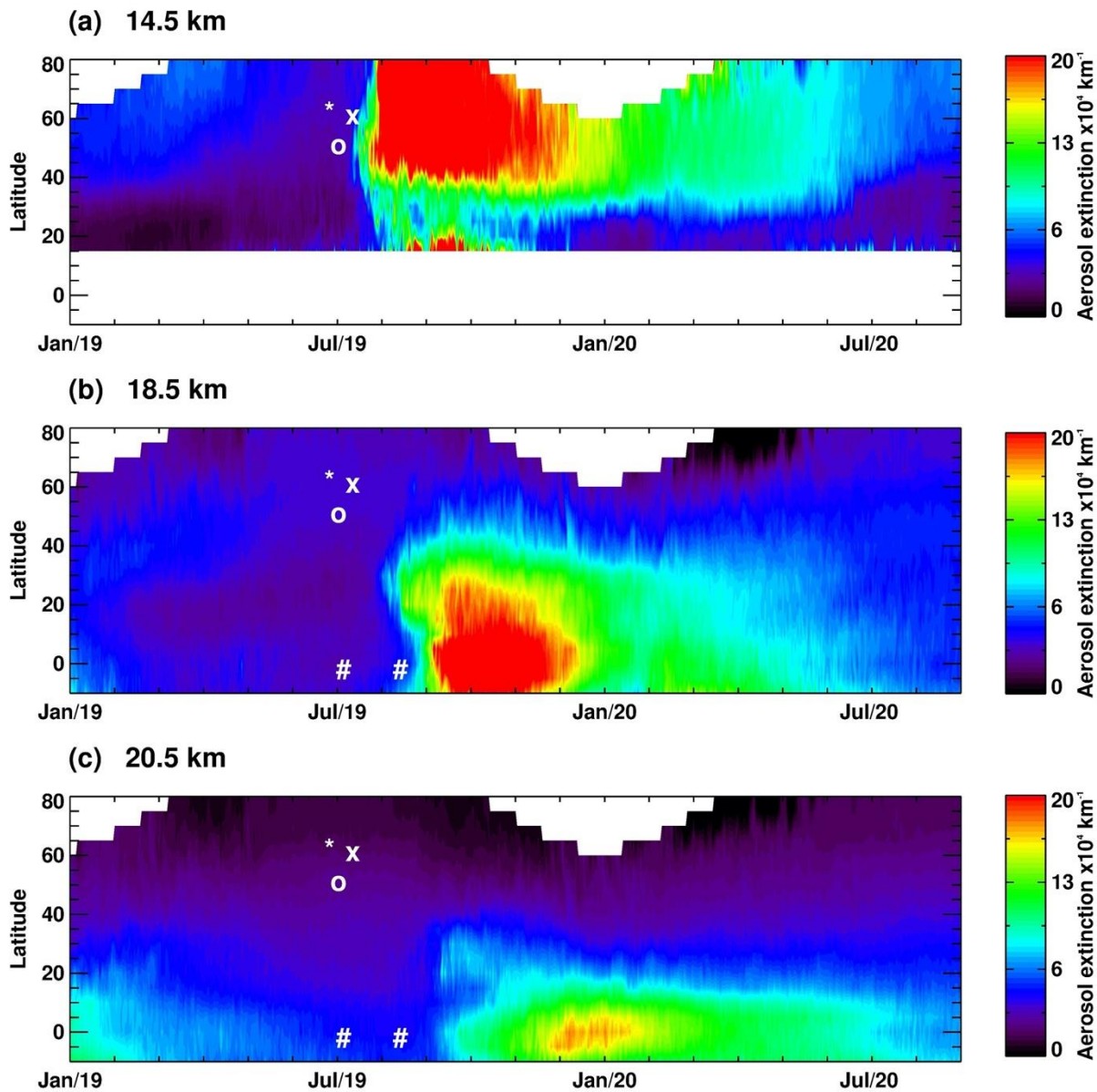

**Figure 7:** Latitudinal distribution of the aerosol extinction coefficient (x $10^4$, km$^{-1}$) at 675 nm at three altitude levels, (a) 14.5 km, (b) 18.5 km, and (c) 20.5 km), averaged every five degrees of latitude. "o" – Raikoke eruption, "*" – Alberta fires, "x" – Siberian fires, "#"– Ulawun eruptions.


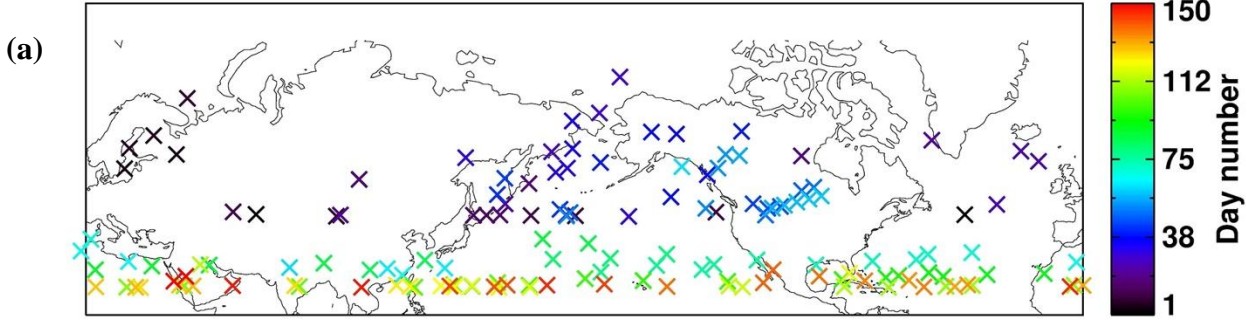

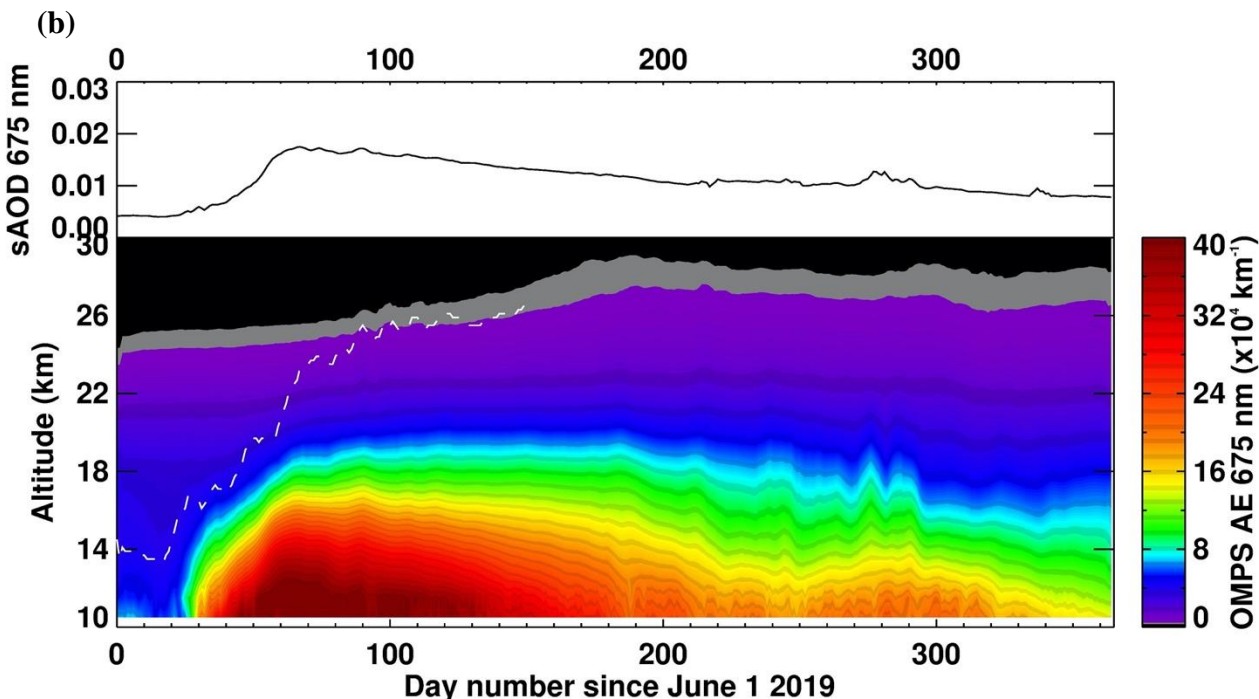

**Figure 8: (a) Daily location of the maximum altitude reached by the Raikoke aerosol plume as detected by OMPS LP, colored by the day number since June 1 and plotted every other day until day 150. Figure 8(b): (bottom) Daily zonal mean aerosol extinction profiles at 675 nm (km$^{-1}$) between 20°S-90°N (which is close to hemispheric coverage) measured by OMPS LP from June 2019 to June 2020 and smoothed spatially using a 5 point boxcar averaging. Only profiles measured above tropopause+1 km are used. The white line is the aerosol plume maximum altitude in km. Figure 8(b): (top) Stratospheric aerosol optical depth (x 10$^3$, sAOD) at 675 nm for latitudes and period similar to Figure 8b. The sAOD is derived by integrating aerosol extinction profiles above the tropopause to 35 km**

Figure 9 shows the altitude-latitude profile of a daily zonal mean ratio of the aerosol extinction coefficient for July 1, 21 and September 1, 30, 2019 to a similar daily zonal mean for a quiet period (June 16-20, 2019) before the eruption. The means were computed without taking into account the "arch effect" (see DeLand and Gorkavyi, 2021, and Sect. 2.2). Therefore, the lower parts of the layers shown may have an overestimated extinction coefficient. In addition to increasing the density of the

background aerosol layer (the contribution of seasonal changes is possible here), an interesting CCC appears at an altitude of about 25 km that will be discussed in Sect. 3.3.

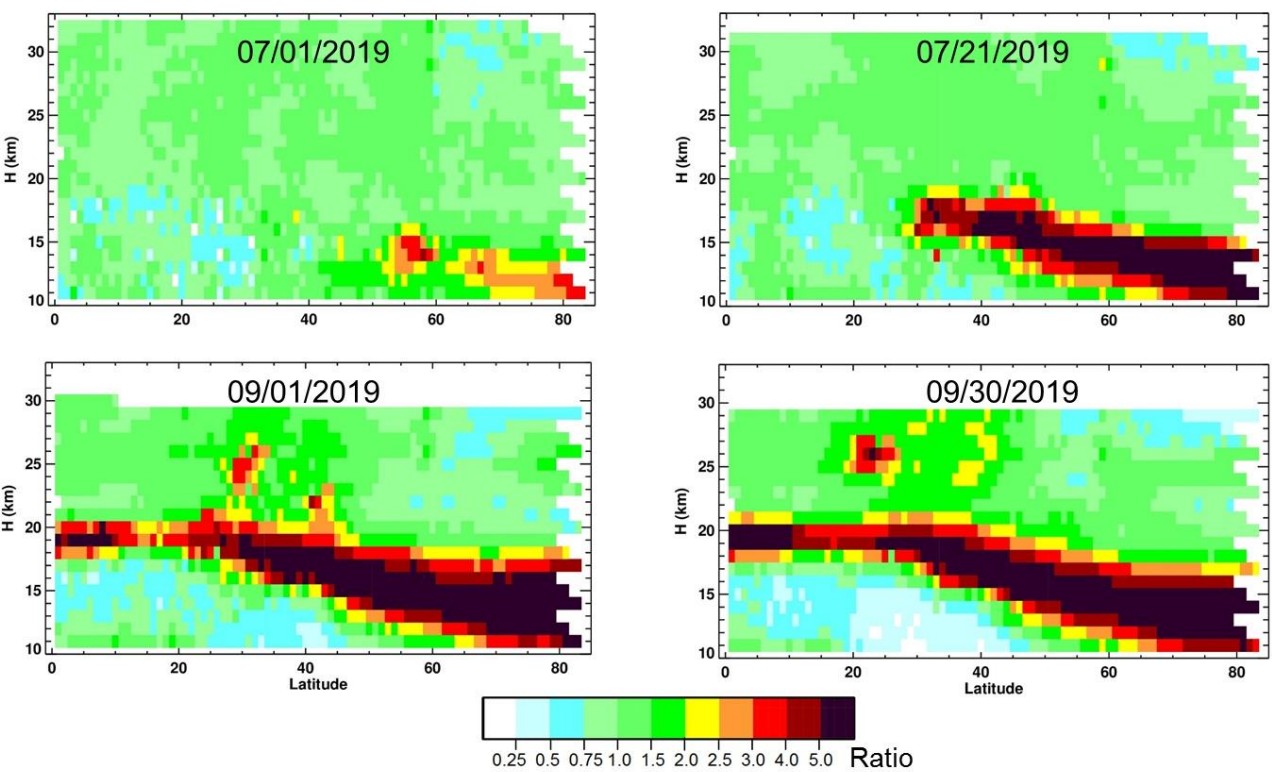

Figure 9: The ratio of the zonal mean, daily average aerosol extinction for the indicated dates after the Raikoke eruption to the averaged aerosol extinction immediately prior to the eruption shows the evolution of the volcanic aerosol in the background aerosol layer (13-20 km) and the occurrence of the CCC near 25 km from OMPS LP data.

The SAGE 1022 nm extinctions for the month of September 2019 are shown in Fig. 10 along with the profile latitudes. Fig. 10a shows that enhanced aerosol extinction extended from the high latitude tropopause to the tropics during September revealing the extent of the aerosol plume. The SAGE aerosol distribution is similar to the OMPS-LP distribution shown in Fig 9. The SAGE III profile latitudes (Fig. 10b) show that SAGE had reasonable coverage from 60°S to 60°N during the month. SAGE III observed the high altitude portion of the Raikoke plume (red circle) at 30°N as the occultations were moving southward. The latitudes for this observation are indicated with the red circle in Fig. 10b. The location of this detached plume is indicated in Fig. 10a which agrees well with the OMPS-LP observations shown in Figs. 4 and 9.

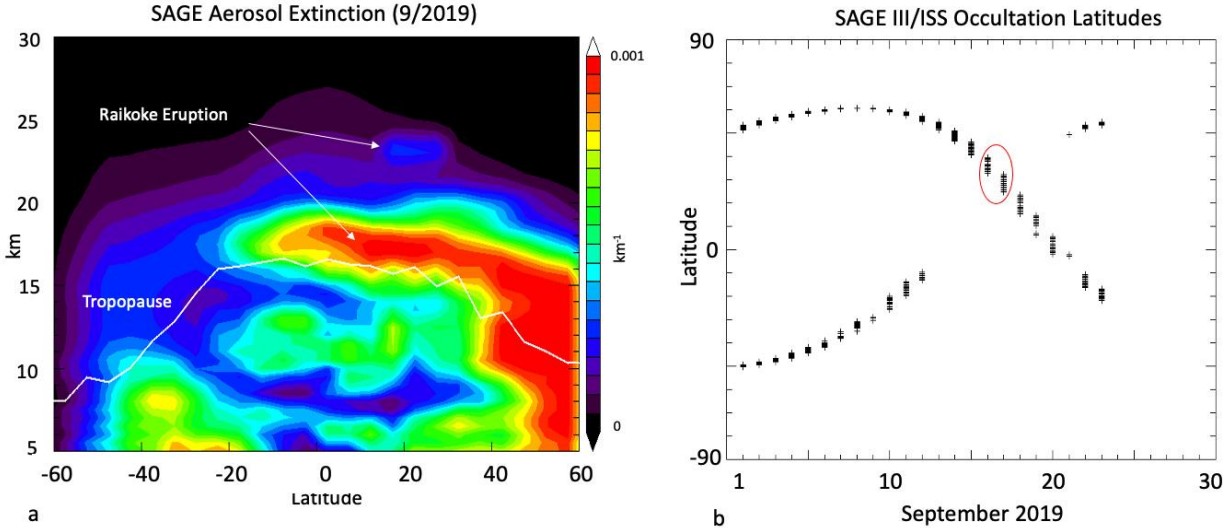

a

b

Figure 10: (a) SAGE aerosol extinction at 1022 nm showing enhanced extinction spreading from high northern latitudes toward the tropics. The elevated part of the plume is indicated with upper arrow. (b) SAGE profile locations during September 2019. Red circle shows where SAGE crossed the elevated portion of the Raikoke plume.

Figure 11 shows three LP aerosol extinction profiles that were measured by OMPS LP on 31 August 2019, more than two months after the Raikoke eruption. The extinction peak at 22-25 km represents the remaining signal of the stratospheric plume from the eruption. The apparent altitude variation between these peaks also shows the impact of the LP viewing geometry on the retrieved extinction profile. Each measured profile (also termed "event") gives an along-track separation of ~125 km. The along-track field of view integrates over a distance of ~180 km for each 1 km vertical sample (DeLand and Gorkavyi, 2021). The OMPS LP version 1.5 algorithm has a restriction on how much the retrieved aerosol extinction is allowed to grow per iteration at each altitude relative to the first guess, which may cause an underestimation of the retrieved aerosol where the plume is concentrated (README Document for the Suomi-NPP OMPS LP L2 AER675 Daily Product, 2019).

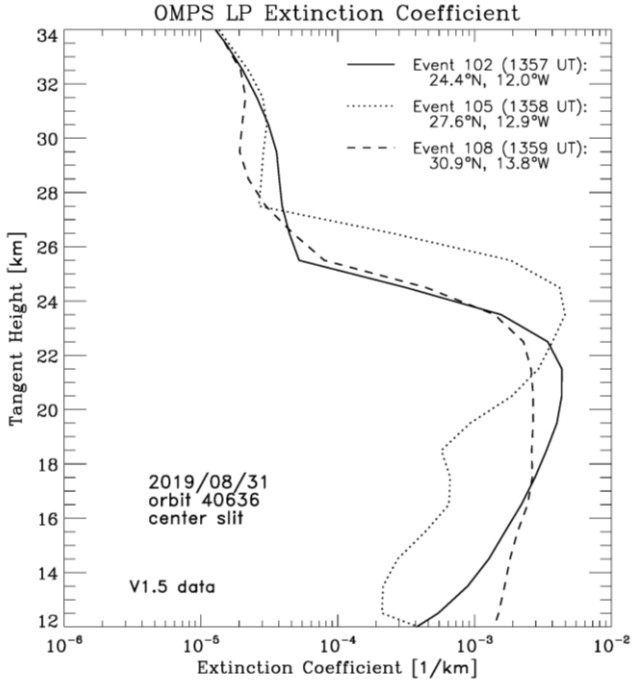

**Figure 11: The aerosol formed as a result of the Raikoke eruption is clearly visible in the extinction coefficient profiles obtained by the OMPS LP aerosol extinction vertical profile months after the eruption.**

### 3.3 Coherent circular cloud (CCC)

The CCC is part of the original aerosol/$SO_2$ cloud resulting from the eruption of the Raikoke volcano. It formed one month after the eruption in the same area where the volcano is located (Fig. 12). Perhaps, the formation of the CCC is associated with a tropospheric vortex, which was observed at the same time in the area (Fig. 12). This vortex could have affected the temperature or pressure in the stratosphere, which caused the CCC. On July 24, 2019, as the CALIOP backscatter data show, the CCC had a height of 19-20 km and quickly shifted to the south (Fig. 12). On July 30, 2019, it reached a latitude of 30°N

and was extended across China, fell into easterly winds and moved west at about 17 m/s (Fig. 13). On August 4, it was recorded over the Persian Gulf region, on August 5 over Egypt, and on August 8 near the Azores.

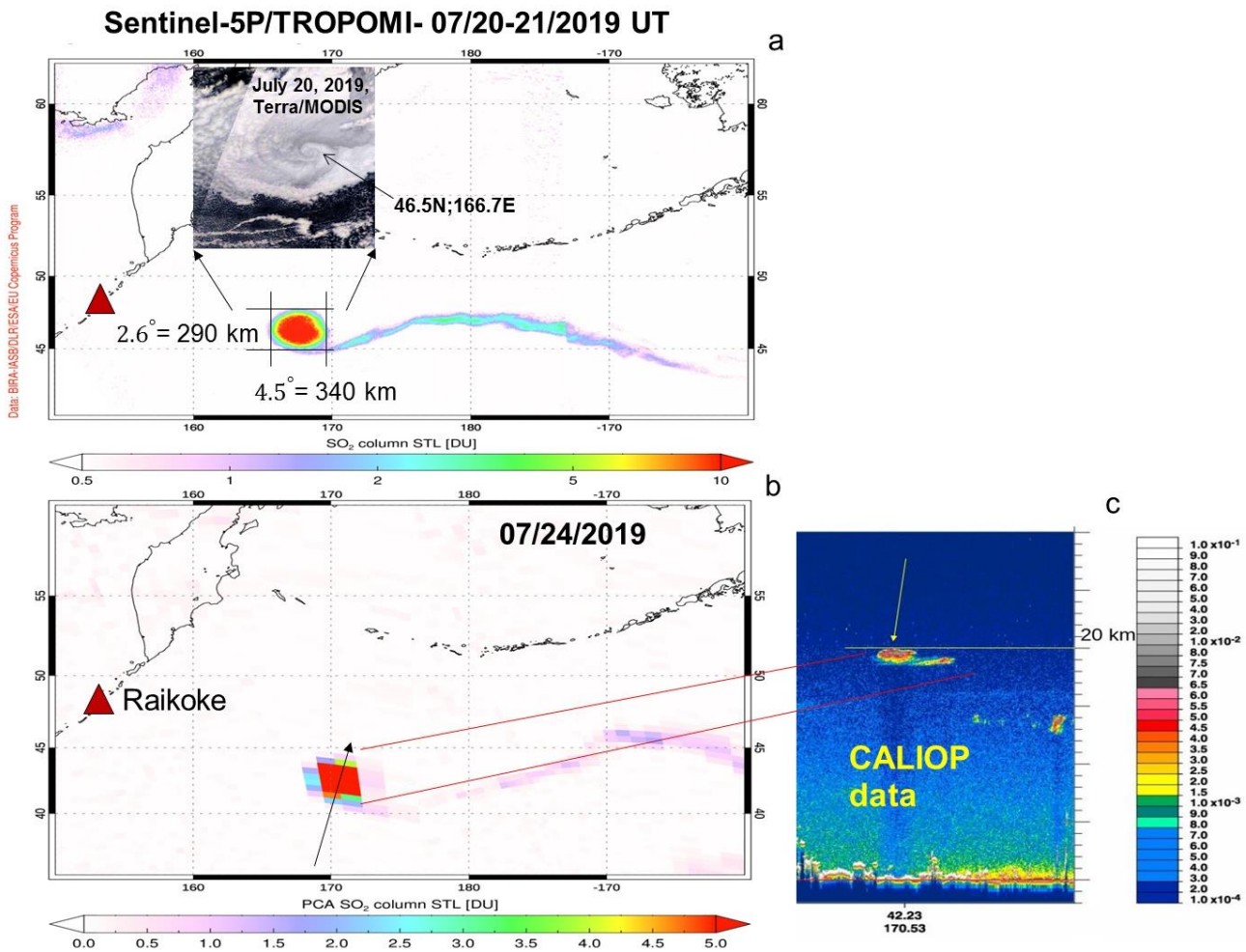

**Figure 12: a)** TROPOMI SO$_2$ STL columns (Upper tropospheric and Stratospheric SO$_2$ column with the center of mass altitude 17 km) for July 20-21 (UTC) plotted using a log scale. Only Raikoke volcano is marked (triangle); **b)** OMPS SO$_2$ STL columns plotted using linear scale for 07/24/2019. Raikoke volcano is marked (triangle). Figures show the formation of CCC, which appeared near Kamchatka and shifted to latitude ~ 30°N and moved to the west at a speed of 1400 km per day. From OMPS NM data we estimated the initial mass of SO$_2$ in this cloud to be ~20 kt; **c)** CALIOP total attenuated backscatter data (km$^{-1}$ sr$^{-1}$) reveal the height of the CCC to be 18.5-20 km.



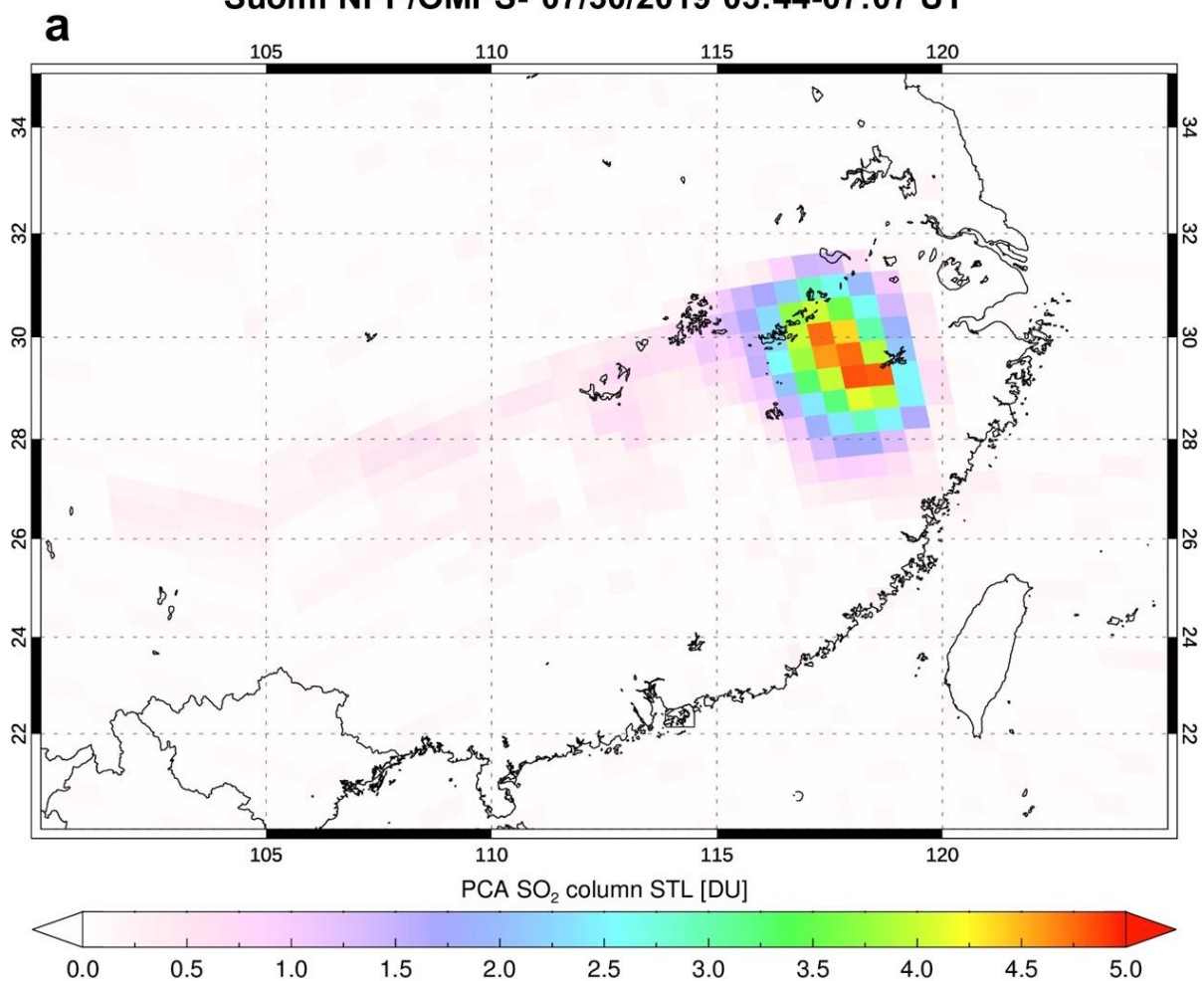

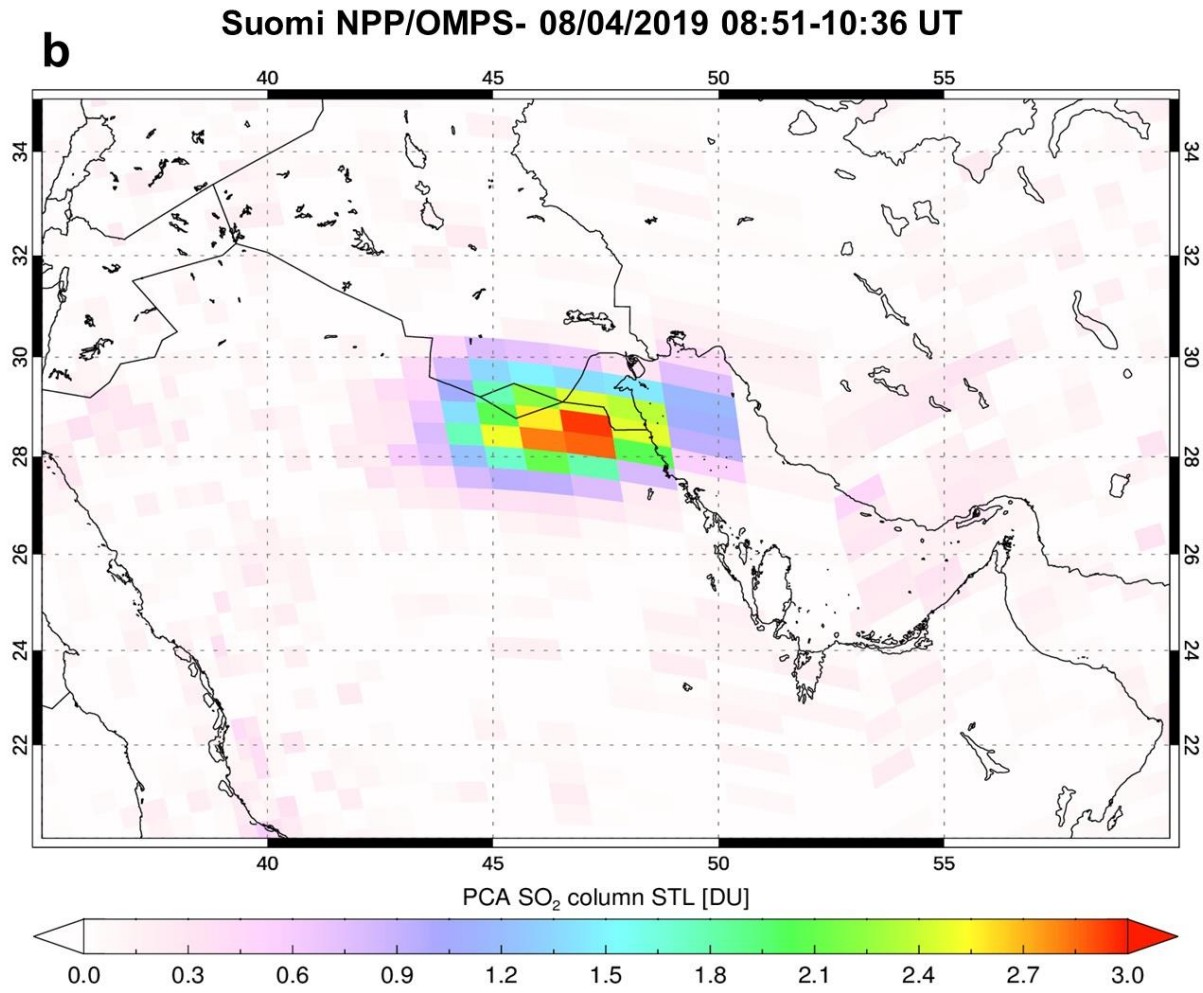

**Suomi NPP/OMPS- 08/04/2019 08:51-10:36 UT**

PCA SO$_2$ column STL [DU]

**Figure 13: OMPS SO$_2$ STL columns plotted using linear scale for 7/30/2019 (top) and 08/04/2019 (bottom). The data show the westward drift of the CCC with easterly stratospheric winds at ~ 14-15°/day, or ~ 1400 km/day, ~ 60 km/h.**

Using the data of CALIOP, OMPS LP and NM as well as OMI, the CCC was tracked until the end of August, while making 1.5 orbits around the Earth. CALIOP and OMPS LP (see Fig. 14 with CALIOP data from August 18 and September 1, 2019 410   and Figs. 4 and 10 from August 31, 2019 – OMPS LP data) were able to track the movement of the accompanying aerosol part of this CCC up to September 22 as it made ~ 3 orbits around the Earth.

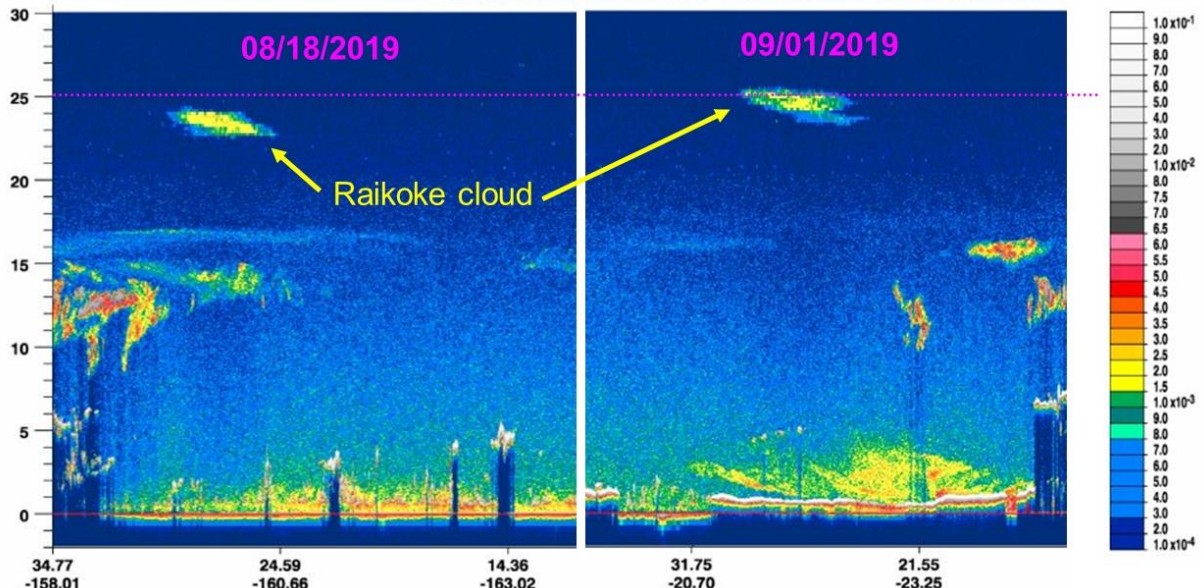

**Figure 14: The CALIOP total attenuated backscatter data (km$^{-1}$ sr$^{-1}$) for 08/18 and 09/01/19, where the CCC is visible at the indicated latitudes and heights.**

Figure 15 shows satellite observations of the CCC from July 18 to September 22, 2019. During this period, the height of the CVC has increased significantly. There are more points by the end of the CCC observation period, because the cloud spreads out and is more often observed by CALIOP. Figure 15 was published in December 2019 at AGU-2019 (Gorkavyi et al., 2019). On September 24, 2019 the CCC was observed by lidars in Hawaii (Chouza et al., 2020). Chouza et al. (2020) traced the trajectory of this cloud back to July 17, 2019. Although the two studies were done independently, they came up with very similar results. Chouza et al. (2020) consider this cloud as a Raikoke plume, but we prefer to call it CCC because it is a very small part of the Raikoke plume.

Figure 16 shows the results of backward trajectory modeling of the CCC for the one month period with August 18 through September 19, 2019. The basic idea is to use the trajectories to see if this long-lived CCC can be correctly transported using assimilated meteorological observations.

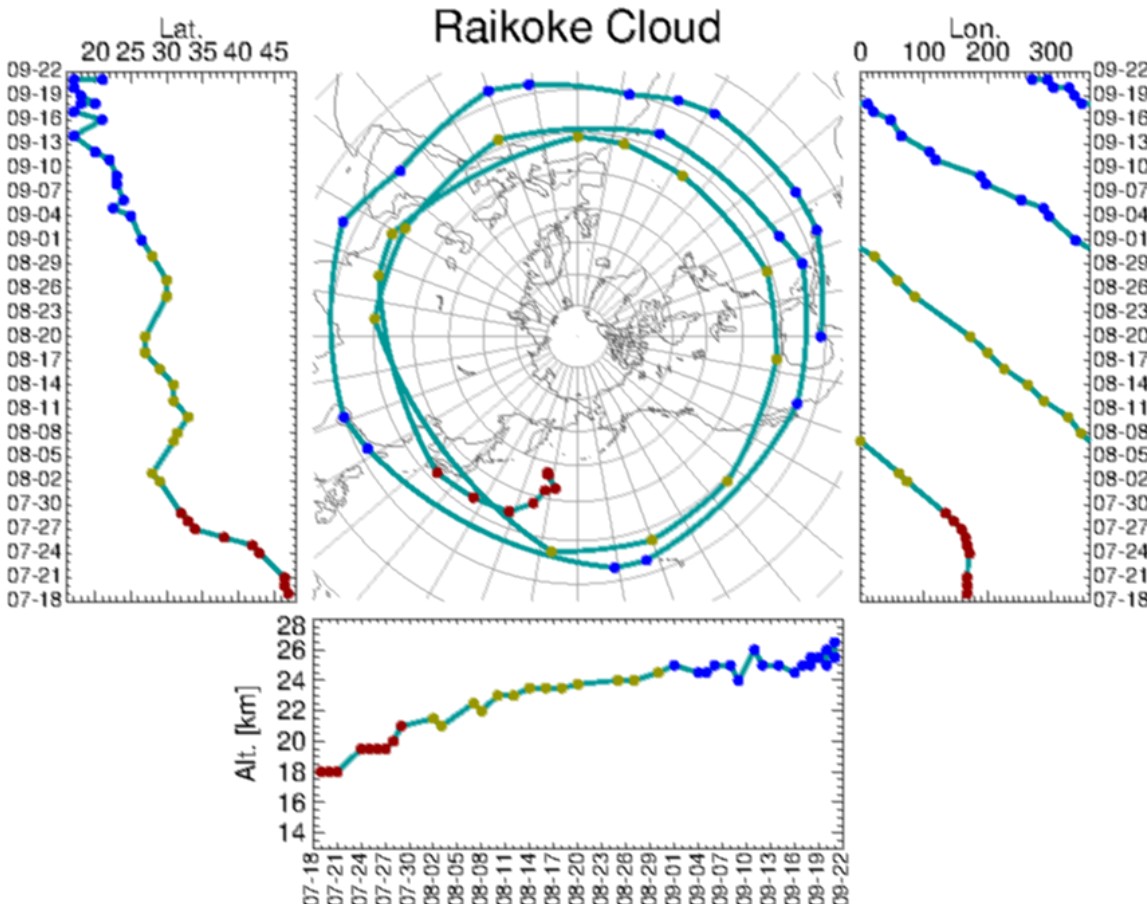

**Figure 15: The observed path of the CCC as inferred from OMI/OMPS/TROPOMI and CALIOP observations. The individual observations are shown by dots color-coded by month.**

We have two hypotheses to evaluate. The first hypothesis ("the great red spot" hypothesis) is that the observed CCC is contained within a gyre (i.e., a whirlpool) that existed from around the time of the eruption on June 21 to September 1 or even later, advected around the globe by the large-scale circulation. The second hypothesis (the "dead fish" hypothesis) is that the CCC was simply advected around by the large-scale winds without being sheared apart because the environment of the summer stratosphere is fairly quiescent.

We inspected the meteorological fields from the NASA's Global Modeling and Assimilation Office (GMAO) Modern-Era Retrospective analysis for Research and Applications, version 2 (MERRA-2) reanalysis (Gelaro et al. 2017) and from the Global Data Assimilation System (GDAS) by the National Centers for Environmental Prediction (NCEP) at the location and altitude of the observed volcanic CCC on July 21 and 24; the winds show no discernible gyre. This is to be expected, as the satellite data that go into these assimilation products are unlikely to pick up features on sufficiently small scales that they would show up clearly and prominently in the winds. Therefore, it is not possible to test the "great red spot" hypothesis using a trajectory model with the wind products available to us.

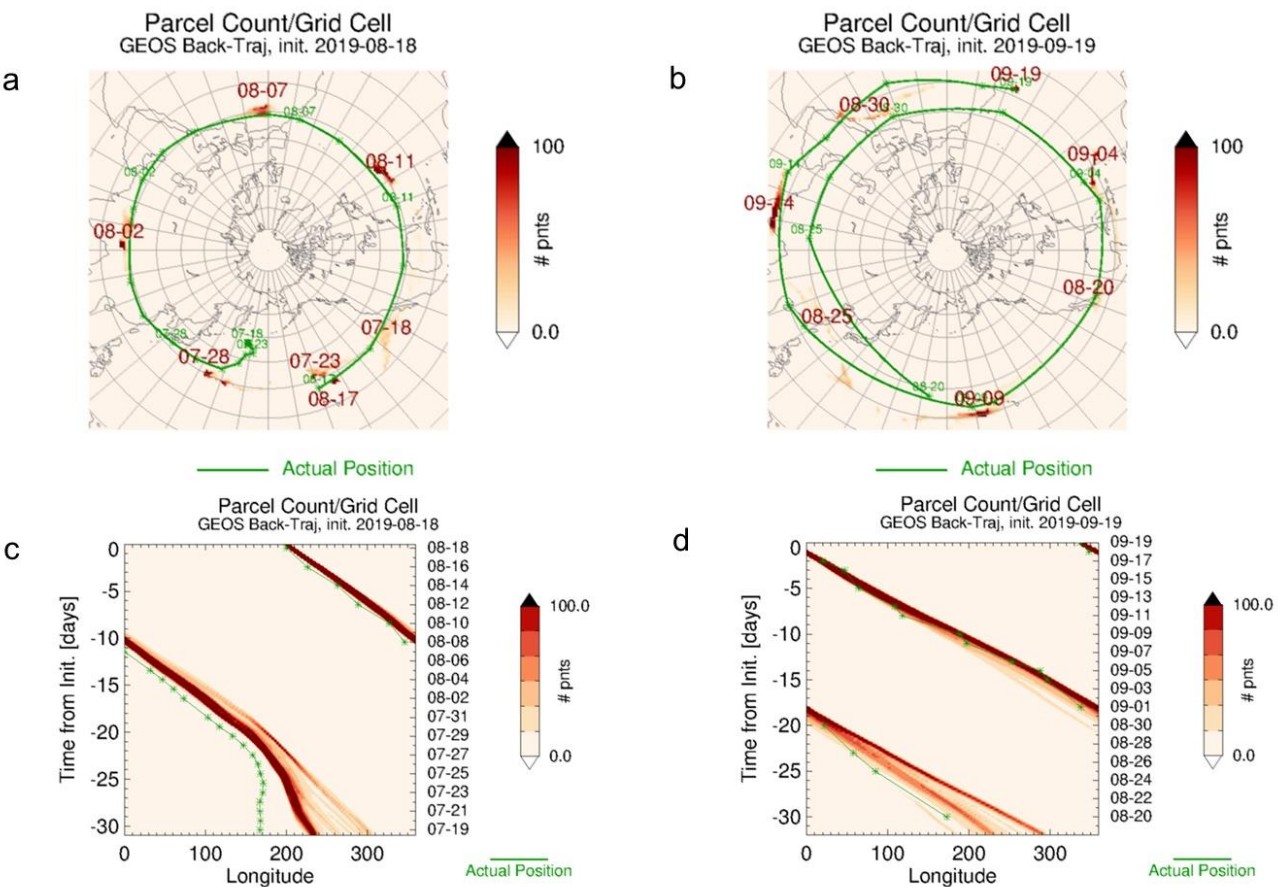

**Figure 16: Backward trajectory modeling: a)** This panel shows several discrete time snapshots from the Goddard "ftraj" back-trajectory modeling for 3000 parcels within a 150 km radius of the observed CCC position on initialized on 2019-08-18; **b)** The trajectory model run was initialized with the observed cloud position on 2019-09-19 and then run backwards to 2019-08-18; **c)** This shows the same model results as for a) but using a different visualization. Here the parcels are binned and counted in longitude and time, regardless of altitude or latitude; **d)** Similar to c), except that the model run is initialized at 2019-09-19 and ran backwards to 2019-08-18.

We can examine the "dead fish" hypothesis. If a trajectory model run shows a cloud of parcels maintaining its integrity over a long period of time, then this hypothesis becomes plausible. There are difficulties, however. First, the winds used to drive the trajectory model have spatial and temporal resolutions that are coarse enough that trajectories cannot be modelled with perfect accuracy. One can expect that a cloud of parcels will lose many of its parcels along the way. It would be unrealistic to expect to duplicate the volcanic cloud's integrity and position as a function of time. Instead, the test is whether the simulated cloud of parcels can maintain its existence over the course of a month or more during this time period. Second, a more serious problem is that of altitude. We can expect a volcanic CCC to experience self-lofting, and the CCC indeed is observed to increase in altitude with time. However, the winds vary with altitude and thus the parcel trajectories will depend critically on getting their altitude changes right.

A carbonaceous aerosol plume associated with wild fires in British Columbia in August 2017 reached the stratosphere a few days following the initial injection and resulting from self-lofting triggered by solar heating (Torres et al., 2020; see similar

effect for Australian fires - Khaykin et al., 2020). The case of a volcanic CCC, which we are considering, is interesting in that the effect of an increase in height from 19 to 26 km is observed for a cloud presumably consisting of sulfate aerosol. In this case, the main increase in the height of the CCC occurred during its movement at approximately the same latitude. Early on (July 19) it is at 80 hPa (450 K), but by September 1 it had risen to 30 hPa (590 K). This is a diabatic self-lofting heating rate of about 3 K per day. Note that this CCC, increasing its height, simultaneously contained a detectable amount of $SO_2$, which is also confirmed by independent satellite data from the Atmospheric Chemistry Experiment - Fourier Transform Spectrometer (ACE-FTS)  (Cameron et al., 2021).

The thermodynamic processes within the cloud are not well-known, so rather than attempting to model the cloud's self-lofting and the overall vertical advection, we impose an approximation to the cloud's vertical motion. We determined an average daily $\theta$ adjustment that would map the column of parcels from its initial $\theta$ range to its final range.  Back-trajectories were then computed isentropically for one day at a time. After each day, the parcels' potential temperature values were changed by their daily adjustment. In this way, the vertical motion of the CCC is guided by observations, while the horizontal motions are unconstrained. Note that this adjustment procedure implies two assumptions: that the self-lofting is linear in time, and that it is linear in $\theta$ as well. The map (see Fig. 16a) displays the parcel population count as a "heat map" or density display: the more intense the color at a given point, the more parcels are near that point (the resolution of the map is $1^\circ$ x$1^\circ$). The plot below the map shows a similar "heat map" of parcel population count as a function of longitude (x-axis) and altitude (y-axis); the resolution here is 1° longitude by 0.5 km altitude.

Figure 16a superimposes several discrete time snapshots from the trajectory model output, starting with the observed cloud position on 2019-08-18. The modelled positions are generally close to the observed, except that the model fails to pick up the effects of the synoptic weather feature beginning around July 23. Figure 16b shows the output from another model run, initialized with the observed CCC position on 2019-09-19 and then run backwards to 2019-08-18. Figure 16c,d shows the same model results but using a Hovmueller-type plot.

The CCC appears not to be a whirlpool, but rather a CCC with easterly flow and drifting ("dead fish" model). Neither the turbulent mixing nor gravity waves penetrate well into the summer stratosphere to mix features, so they tend to move intact if injected at the right time.

**6 Conclusion**

We studied the process of converting $SO_2$ to aerosol for the June 21, 2019 eruption of the Raikoke volcano (Kuril Islands, Russia, 48°N, 153°E). The peak sulfate aerosol extinction is 1.5 months behind the time of the release of $SO_2$ in the Raikoke eruption.  We see two different regimes in Raikoke $SO_2$ clouds with significantly different e-folding times. Initially, the $SO_2$ mass decreases with a longer apparent e-folding lifetime of ~19 days, compared to the later period when the decrease in $SO_2$ mass accelerates with the apparent e-folding lifetime ~8 days. The e-folding lifetime of ~19 days, derived from the earlier stage after the eruption, probably reflects the true conversion of $SO_2$ into sulfate aerosol. After that, the detection limit of the OMPS instrument becomes a more important factor, leading to faster apparent decay of the total observed $SO_2$.

We evaluated the influence of the arch effect on the calculation of the extinction coefficient of a finite volcanic cloud using homogeneous spherical shell atmosphere assumption adopted for processing the data of the OMPS Limb Profiler.  We have shown that this effect is significant; therefore, it should be investigated by more accurately accounting for inhomogeneous atmospheric composition along the line of sight within the framework of a 2D model of radiative transfer.

We examined the unusual coherent circular cloud (CCC) of $SO_2$ and aerosol, which was observed for more than 2 months (from July 18 to September 22). The CCC was embedded in the summer easterly flow in the stratosphere and demonstrated an unprecedented structural cohesion along with diabatic "self-lofting" heating rate of about 3 K per day. For 2 months, the CCC circled the globe almost three times at the latitude of 30°N (during September, the CCC shifted south to 15-20°N) and increased its height from 19 to 26 km.

**Data availability**

The SO$_2$ data for OMPS, OMI and TROPOMI data are available at https://so2.gsfc.nasa.gov/ and Goddard Earth Sciences Data and Information Services Center (GES DISC) at https://disc.gsfc.nasa.gov/datasets/OMPS_NPP_NMSO2_L2_2/summary. CALIOP data are available at https://www-calipso.larc.nasa.gov/products/.
The OMPS-NPP L2 LP Aerosol Extinction Vertical Profile (v1.5) data are available at https://disc.gsfc.nasa.gov/datasets/OMPS_NPP_LP_L2_AER675_DAILY_1.5/summary (Bhartia and Torres, 2019).
The SAGE III/ISS data are available at https://asdc.larc.nasa.gov/project/SAGE%20III-ISS (John M. Kusterer)

**Competing interests**

The authors declare that they have no conflicts of interest.

**Authors's contributions**

NG, CL, LL, SC, MD, MS, GT developed computer codes and algorithms, analyzed the results and wrote the manuscript. NK, JJ, AV set the task of developing, supported the development of the algorithm, analyzed the results and wrote the manuscript. PN, OT, PC analyzed the data and results and wrote the manuscript.

**Financial support**

This work was supported by the NASA Aura project (OMI core team) managed by Ken Jucks. The SNPP/OMPS SO$_2$ product has been developed with support from the NASA Science of Terra, Aqua, and Suomi NPP program (grant # 80NSSC18K0688) managed by Barry Lefer. OMPS LP aerosol data were produced with support from NASA contract # NNH17HP01C. G. Taha is supported by the National Aeronautics and Space Administration grant # 80NSSC18K0847. Nickolay Krotkov and Can Li acknowledge support by the NASA Applied Sciences Disasters program managed by David Green and John Murray.

**Acknowledgements**

The authors thank the OMPS, OMI, TROPOMI, MLS and CALIOP teams for providing the OMPS, OMI, TROPOMI, MLS and CALIOP data presented, respectively. The authors are grateful to M. Fromm and an anonymous referee for useful comments and remarks.

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
