# Peer review of "Tracking aerosols and SO2 clouds from the Raikoke eruption: 3D view from satellite observations"

_Atmospheric Measurement Techniques, 2021_

## Referee Comment (RC1)

**Review of amt-2021-58**

The manuscript studies the dynamics of the stratospheric injection from the Raikoke volcanic eruption. Nadir sensors measuring the $SO_2$ column were combined with a limb sensor measuring stratospheric aerosol extinction to obtain a more complete picture of the volcanic effects. A coherent circular cloud of $SO_2$ and aerosol was identified and studied with a trajectory model. Overall I found the paper well-written, interesting, and relevant for the scientific community as a whole. However I have some major concerns related to some of the analysis of the limb stratospheric aerosol data. I believe that these are fixable, and would recommend that the paper is published after these issues are corrected.

**General Comments**

My main concern is related to that of the "arch effect" correction. The specific comments below go into more detail on my concerns, but the primary one is that a 1D retrieval does not only introduce "arches", it also underestimates the main plume. The underestimation of the main plume is not taken into account by the analysis performed here. I don't believe the correction itself actually influences any of the main results of the paper, so this could easily be changed, but I feel like something should be done. I would suggest either removing it entirely, providing more justification that the correction is in fact something positive, or adjusting the manuscript so that the correction is presented as a potential source of error instead of an actual correction.

**Specific Comments**

**p.4 l.86-98:** At this point various resolutions (wavelength and spatial) are introduced for TROPOMI and OMPS NM. The importance of the spatial resolution is obvious but the importance of the wavelength resolution is not. I would suggest that some characteristics of the retrieval are introduced either in addition or instead, i.e., any available bias/precision estimates for the $SO_2$ column.

**Section 2.2:** This section is missing some discussion of the microphysical assumptions necessary for the OMPS LP aerosol retrieval and how they differ from the CALIOP retrieval. I believe this is a gamma particle size distribution with fixed parameters?

**Figure 3:** After staring at this figure for a while I could not reason out what is actually being shown. To demonstrate the relationship between $H$ and $h$ we would need to see a single cloud, with two different lines of sight/observer locations, but instead we see five different clouds and a few tangent heights? Is it intended that the "five clouds" A-E are not different clouds but the same cloud seen at different times? If so it is also confusing since OMPS LP is backwards looking the first observation is "E" instead of "A".

**p.4 l.106-108:** Are all three slits from OMPS LP used in this analysis or just the central slit?

**p.4 l.118:** "LP signal strength (e.g., extinction coefficient)" is confusing, extinction coefficient may be related to signal strength but it is not an example of signal strength.

**p.5 l.136:** You state that the displacement is approximately equal to a latitude displacement, I assume that is only for illustrative purposes since there is no need to make this approximation in the actual correction?

**p.5 l.141:** "We can therefore use Eq. (1) to calculate and apply a correction for determining the magnitude and position of an aerosol cloud." I understand how the equation can be used to calculation a correction for the position of the aerosol "cloud", but I don't see how it can be used to correct for its magnitude.

**p.6 l.157-161:** I agree this algorithm will remove the "arch" however it is not convincing to me that this improves the aerosol extinction, in fact, I am not even convinced that a 1D retrieval always overestimates the total retrieved optical depth as would be suggested by the text. If you imagine an aerosol point source and do successive 1D retrievals along the orbit you will obtain an arch, just as the authors suggest. The arch is obviously unphysical, and for this reason the authors remove it. But, for the one 1D retrieval where the point source is located at the tangent point, the 1D retrieval will actually greatly underestimate the magnitude of the point source. The reason for this is that the 1D retrieval is assuming horizontal homogeneity, so it cannot add a point source, it must add aerosol with a greater extent. In addition the 1D retrieval will also underestimate below the point source because these altitudes in the 1D forward model contain extra aerosol scattering from assuming horizontal homogeneity. The underestimation effect is completely ignored by the authors and for this reason I do not believe the arch correction as presented is meaningful. I do not see any way to either remove the biases of a 1D retrieval or estimate its effect that does not involve full two-dimensional radiative transfer simulations.

**p.6 l.165-166:** The wording here could give the impression that a tomographic retrieval has never been implemented for OMPS LP aerosol extinction, however it appears it was done in the Zawada et. al. reference on the same line.

**Figure 5:** When I look at this figure it tends to reinforce my belief that the "arch effect" is not doing what is expected. Should the correction not be close to 0 before the effects of the eruption? Here it looks like the presence of a plume has no effect on the "arch effect".

**p.7 l.188:** Is there a reason to only include OMPS NM here and not TROPOMI?

**Figure 5:** What is the cause of the artifact in $SO_2$ at 35 days? If it is a sampling effect I would suggest to remove the datapoint.

**p.9 l.228:** "more and more pixels with SO2 fall below the detection limit of the OMPS NM sensor" Does this mean that only pixels where $SO_2$ is detected are included in the analysis? Presumably if every pixel is included then this would only lead to poorer precision.

**Figure 7:** Specify which tropopause was used here for the integration.

**p.12 l.276:** "The along-track field of view integrates over a distance of $\sim$180 km for each 1 km vertical sample" While true that a 1 km shell ends up having a $\sim$ 180 km horizontal extent geometrically, the actual horizontal resolution of a limb sounder is more complicated than this. See for example, von Clarmann et. al. 2009
von Clarmann, T., De Clercq, C., Ridolfi, M., Höpfner, M., and Lambert, J.-C.: The horizontal resolution of MIPAS, Atmos. Meas. Tech., 2, 47–54, https://doi.org/10.5194/amt-2-47-2009, 2009.

**Technical Corrections**

**Figure 4:** In the caption "For accurately calculating" → "To accurately calculate"

**p.6 l.165:** Livesay → Livesey and Zawanda → Zawada

**p.18 l.369:** "Figure15" → "Figure 15a"

---

## Author Comment (AC2)

**Review of amt-2021-58**

The manuscript studies the dynamics of the stratospheric injection from the Raikoke volcanic eruption. Nadir sensors measuring the SO 2 column were combined with a limb sensor measuring stratospheric aerosol extinction to obtain a more complete picture of the volcanic effects. A coherent circular cloud of SO2 and aerosol was identified and studied with a trajectory model. Overall I found the paper well-written, interesting, and relevant for the scientific community as a whole. However I have some major concerns related to some of the analysis of the limb stratospheric aerosol data. I believe that these are fixable, and would recommend that the paper is published after these issues are corrected.

**General Comments**

My main concern is related to that of the "arch effect" correction. The specific comments below go into more detail on my concerns, but the primary one is that a 1D retrieval does not only introduce "arches", it also underestimates the main plume. The underestimation of the main plume is not taken into account by the analysis performed here. I don't believe the correction itself actually influences any of the main results of the paper, so this could easily be changed, but I feel like something should be done. I would suggest either removing it entirely, providing more justification that the correction is in fact something positive, or adjusting the manuscript so that the correction is presented as a potential source of error instead of an actual correction.

ANSWER 1:  We agree that the density of the local cloud, calculated within the 1D RTM, may differ significantly from the real one. But at the same time it is clear that all values of the cloud density that are obtained below its real height are artifact and must be removed. We improved Figure 3 (now 3a) and made an additional Figure 3b. We have expanded our discussion of the arc effect considerably. We also added the following text to the article: "We consider this procedure of correction only as an estimate, which shows the possible significance of the arch effect".  "These estimates show the significance of the arch effect.  For more accurate calculations, the arch effect should be investigated within a 2D RTM".

**Specific Comments**

p.4 l.86-98: At this point various resolutions (wavelength and spatial) are introduced for TROPOMI and OMPS NM. The importance of the spatial resolution is obvious but the importance of the wavelength resolution is not. I would suggest that some characteristics of the retrieval are introduced either in addition or instead, i.e., any available bias/precision estimates for the SO 2 column.

ANSWER 2: We added next statement:  "For large volcanic $SO_2$ signals like Raikoke, comparisons between TROPOMI and SNPP/OMPS so far (for several eruptions) show little bias, with the total $SO_2$ mass estimates from the two normally agreeing to within 5-10% (with the exception of the very early stages of large eruptions, where the density of SO2 and/or volcanic ash is too high to be fully accounted for in operational algorithms). For retrieval noise on a pixel-to-pixel basis, SNPP/OMPS $SO_2$ (for stratospheric clouds) is less than 0.1 DU. TROPOMI's noise on a pixel-by-pixel base is several times greater, but once TROPOMI pixels are averaged to OMPS footprints, the noise is reduced by ~30%."

**Section 2.2**: This section is missing some discussion of the microphysical assumptions necessary for the OMPS LP aerosol retrieval and how they differ from the CALIOP retrieval. I believe this is a gamma particle size distribution with fixed parameters?

ANSWER 3: In this paper, we only used images of CALIOP total attenuated backscatter signal as a reference to identify the Raikoke plume location and altitude. We didn't use any retrieved aerosol data. We have added the following to L101 (first line in 2.2): "The OMPS LP V1.5 aerosol retrieval algorithm is described Sects. 2 and 3 of Chen et al. (2018)." We also replaced "CALIOP aerosol data" in L286 (second line in 3.3) with "CALIOP backscatter data".

**Figure 3:** After staring at this figure for a while I could not reason out what is actually being shown. To demonstrate the relationship between H and h we would need to see a single cloud, with two different lines of sight/observer locations, but instead we see five different clouds and a few tangent heights? Is it intended that the "five clouds" A-E are not different clouds but the same cloud seen at different times? If so it is also confusing since OMPS LP is backwards looking the first observation is "E" instead of "A".

ANSWER 4: We have significantly improved Figure 3 and expanded the explanation for it. We consider the positions A-E as different positions of the same cloud. The order of the letters has changed.

**p.4 l.106-108**: Are all three slits from OMPS LP used in this analysis or just the central slit?

ANSWER 5: For Figs 4 and 11 only central slit was used, for other cases – all 3 slits.

**p.4 l.118**: "LP signal strength (e.g., extinction coefficient)" is confusing, extinction coefficient may be related to signal strength but it is not an example of signal strength.

ANSWER 5: The text has been changed: "LP (e.g., extinction coefficient)"

**p.5 l.136:** You state that the displacement is approximately equal to a latitude displacement, I assume that is only for illustrative purposes since there is no need to make this approximation in the actual correction?

ANSWER 6: Correct.

**1p.5 l.141:** "We can therefore use Eq. (1) to calculate and apply a correction for determining the magnitude and position of an aerosol cloud." I understand how the equation can be used to calculation a correction for the position of the aerosol "cloud", but I don't see how it can be used to correct for its magnitude.

ANSWER 7: We added: "we believe that all parts of the arch below the real height are artifacts, so the value of the extinction coefficient for them should be equal to zero".

**p.6 l.157-161:** I agree this algorithm will remove the "arch" however it is not convincing to me that this improves the aerosol extinction, in fact, I am not even convinced that a 1D retrieval always overestimates the total retrieved optical depth as would be suggested by the text. If you imagine an aerosol point source and do successive 1D retrievals along the orbit you will obtain an arch, just as the authors suggest. The arch is obviously unphysical, and for this reason the authors remove it. But, for the one 1D retrieval where the point source is located at the tangent point, the 1D retrieval will actually greatly underestimate the magnitude of the point source. The reason for this is that the 1D retrieval is assuming horizontal homogeneity, so it cannot add a point source, it must add aerosol with a greater extent. In addition the 1D retrieval will

also underestimate below the point source because these altitudes in the 1D forward model contain extra aerosol scattering from assuming horizontal homogeneity. The underestimation effect is completely ignored by the authors and for this reason I do not believe the arch correction as presented is meaningful. I do not see any way to either remove the biases of a 1D retrieval or estimate its effect that does not involve full two-dimensional radiative transfer simulations.

ANSWER 8:  We improved Figure 3 and made an additional Figure 3b, as well as discussed in more detail the problems of limb observation. Also we added new statements (see ANSWER 1).

**p.6 l.165-166:** The wording here could give the impression that a tomographic retrieval has never been implemented for OMPS LP aerosol extinction, however it appears it was done in the Zawada et. al. reference on the same line.

ANSWER 9:  The text has been changed:  "One way to account for such effects is to use a two-dimensional (2D) radiative transfer model (RTM) that is able to account for such effects along with multiple observations in a tomographic retrieval (e.g., Livesey et al., 2006; Zawada et al., 2018; Loughman et al., 2018). Instead, we have developed an a posteriori adjustment method…"

**Figure 5:** When I look at this figure it tends to reinforce my belief that the "arch effect" is not doing what is expected. Should the correction not be close to 0 before the effects of the eruption? Here it looks like the presence of a plume has no effect on the "arch effect".

ANSWER 10:  See ANSWER 8.

**p.7 l.188:** Is there a reason to only include OMPS NM here and not TROPOMI?

ANSWER 11:  TROPOMI, unlike OMPS, does not have a limb sensor.

**Figure 5:** What is the cause of the artifact in SO 2 at 35 days? If it is a sampling effect I would suggest to remove the datapoint.

ANSWER 12: We have corrected Figure 5.

**p.9 l.228:** "more and more pixels with SO2 fall below the detection limit of the OMPS NM sensor" Does this mean that only pixels where SO 2 is detected are included in the analysis? Presumably if every pixel is included then this would only lead to poorer precision.

ANSWER 13: Correct. Only pixels where SO2 is detected are taken into account in the analysis of SO2 dynamics. Also we added in the text:  "(OMPS  in the stratosphere can typically detect 0.2-0.3 DU of SO2)"

**Figure 7:** Specify which tropopause was used here for the integration.

ANSWER 14:  We have changed Figure 7.

**p.12 l.276:** "The along-track field of view integrates over a distance of ~180 km for each 1 km vertical sample" While true that a 1 km shell ends up having a ~ 180 km horizontal extent geometrically, the actual horizontal resolution of a limb sounder is more complicated than this. See for example, von Clarmann et. al. 2009 (von Clarmann, T., De Clercq, C., Ridolfi, M., H¨opfner, M., and Lambert, J.-C.: The horizontal resolution of MIPAS, Atmos. Meas. Tech., 2, 47–54, https://doi.org/10.5194/amt-2-47-2009).

ANSWER 15:  Correct. See ANSWER 8.

**Technical Corrections**

Figure 4: In the caption "For accurately calculating" → "To accurately calculate"

**p.6 l.165:** Livesay → Livesey and Zawanda → Zawada

**p.18 l.369:** "Figure15" → "Figure 15a"

ANSWER 16:  We have fixed all the mentioned errors.

---

## Author Comment (AC3)

Gorkavyi et al. (hereafter "Gteam") have three primary thrusts in this manuscript on the Raikoke volcano plume of 2019, to 1. characterize the stratospheric plume SO2 and sulfate evolution, 2. Introduce a limb-view retrieval artifact called the "arch effect," and 3. Follow a compact plume element termed a coherent circular cloud (CCC). These themes are carried out by invoking five different satellite-based measurement/retrieval products, three nadir-imaging and two vertical profiling.

The above themes, data, and methods are an appropriate fit for AMT.  It is evident from the data presented by Gteam that the Raikoke volcanic cloud is scientifically important, even remarkable, and needs to be thoroughly characterized in the literature. Moreover, Gteam present a novel method for limb-view aerosol profile handling to deal with a known limitation in the limb approach to quantifying perturbations that are inhomogeneous and/or geographically small in the context of instrument resolution.

My summary assessment of GTeam is that each of the three thrusts need substantial revision to qualify as publishable. In each respect, the material has either been presented in prior literature and not fully recognized herein, or the current presentation lacks clarity, validity, or motivation. Details of these primary concerns are next. Following these is a list of

minor concerns. Technical concerns and questions are handled by comments and annotations to the manuscript, provided separately.

Primary Concerns

Gteam's foremost new content regards the "arch effect." Here they focus on a fundamental uncertainty with respect to limb-view spectroscopy, that being the accuracy of retrieving extinction and altitude of an object that is small with respect to the instrument's (OMPS-LP in this case) line-of-sight field of view (FOV). The problem common to all such stratospheric limb viewers is that they peer through a roughly 200 km atmospheric path length (~180 km quoted by Gteam for OMPS-LP) composed of a wide altitude range in the Earth-centric reference frame. Generally, the assumption is made that any object in the view path occupies the entire FOV. To the extent that any object (like a meteorological cloud or aerosol plume) is smaller (vertically or horizontally) than the instrument's FOV, one or both of the retrieved extinction and feature altitude will be biased low. In highlighting the "arch effect," Gteam seem to be suggesting that the Raikoke sulfate plume presents a source of systematic error in OMPS-LP extinction and plume-height results. Hence the motivation for their focus on an adjustment to mitigate the "arch effect."

The "arch effect" argument, to be persuasive, requires at least one additional data item independent of OMPS LP to

characterize cloud- or plume-object height, vertical thickness, and horizontal extent all within the OMPS-LP FOV. I.e. OMPS-LP on its own is under-constrained for such an assessment. Gteam present the arch effect argument by invoking only OMPS-LP profile data. Moreover, the illustrative example of the arch effect is for a scene that has a Raikoke plume object that is ~40 days old. Even though it is presumed to be compact with respect to the OMPS LP FOV, this (and most other ) Raikoke plume elements have spread over distances greatly exceeding the OMPS LP FOV. With the data that Gteam present as an example of the arch effect—a consecutive sequence of OMPS LP aerosol profiles—it is not possible to know the true size of the Raikoke plume object precisely in the FOV. It is known, however, that the CCC the Gteam is following has horizontal dimensions as great as ~600 km, based on the CALIOP curtain one day later, shown in Figure 13. It is conceivable that the 31 August OMPS example given for the arch effect was a case of the limb view sampling a peripheral part of the CCC, presenting a much smaller horizontal distance. But insufficient independent information is provided to inform the reader.

Exploring complementary data sets for the Gteam 31 August 2019 case of the arch effect, it becomes apparent that indeed the OMPS center slit sampled CCC they are tracking.  The GOES East visible reflectance image shown below illuminates the CCC over coastal northwest Africa at OMPS measurement time:

~13:50 UTC. The visible meridional extent of the CCC is ~3.4 deg. Latitude, approximately 380 km. This is roughly doule the OMPS tangent path FOV.

[Figure]

The NASA micropulse lidar (mplnet.gsfc.nasa.gov) at Santa Cruz, Tenerife, (due west of the OMPS extinction-profile curtain in Figure 4) measured the CCC as it blew west.  A snapshot of the normalized backscatter ratio profile at 22UTC 31 August is below.

[Figure]

MPLNET Santa_Cruz_Tenerife 2019-08-31 22:10:30 V3_L1_NRB: NRB

Energy: 6.518 uJ  (Set Point: 6.244 uJ)
Box Temperature: 23.883 C  (Set Point: 23.380 C)
Detector Temperature: 26.841 C  (Set Point: NaN C)
Laser Temperature: 29.920 C  (Set Point: NaN C)
Solar Background: 4.320E-05 MHz

QA Flag: Good

Back trajectories to OMPS time 31 August show that the plume over Tenerife indeed passed over  the OMPS curtain at OMPS measurement time.

[Figure]

The nadir GOES image and the precise lidar profile tightly constrain the plume geometry encountered by OMPS. Thus it appears that the CCC conditions encountered by OMPS are

vastly inconsistent with the limited horizontal cloud extent used to illustrate and motivate the arch effect principle. See the annotations to Gteam's schematic in the separate manuscript.

Given these two additional views of the CCC encountered by OMPS LP, it is largely uncertain as to how to interpret the OMPS extinction profiles in Figures 10 and 4. This does not appear to be a candidate for the arch effect, and the maximum extinction in the CCC—according to the example of it shown in the 1 September CALIOP curtain (Figure 13)—may be as much as an order of magnitude greater than retrieved by OMPS (CALIOP backscatter x lidar ratio of 50 sr). Considering all these factors, Gteam is encouraged to either clarify and bolster their arguments for this arch effect example, or find another case where it can be independently shown that a sub-FOV-filling plume element is sampled by OMPS.

If such an example can be demonstrated, it still seems unlikely that the aging/spreading Raikoke plume as sampled by OMPS LP justifies wholistic application of the arch-effect adjustment. The shear variety and complexity of plume presentations (e.g. the various CALIOP curtains illustrated by Gteam) to OMPS during the analysis time frame indicate that compact-to-the-point-of-sub-FOV plume elements are rare. This includes the CCC and the meridionally broader Raikoke layers at lower altitudes do not meet the size-limited view as depicted in Figure 3.

It is essential to characterize the schematic angles and "object" (also referred to as a "cloud") in terms of their geophysical horizontal and vertical dimensions. It is all important to know what size range of cloud in the along-track FOV direction creates the arch effect. It is difficult to know from the schematic how realistic the cloud object is, but it appears to be tiny in relation to the 180 km OMPS LP FOV.

To make a compelling argument for the arch effect, three things would have to be presented that are lacking. 1. A case study involving an independent plume-object physically characterized, 2. Such an element being located within the OMPS LP FOV (i.e. a space-time match), and 3. The plume element being demonstrably smaller than the FOV (as drawn schematically in Figure 3). An example of such a case study is provided in Penning de Vries et al. (2014) (cited by Gteam), who attempted to reconcile SCIAMACHY limb-scatter profiles of the Nabro volcano stratospheric plume with simultaneous nadir $SO_2$ imagery. It is notable that the Nabro example was when the plume was less than two days old; demonstrably compact.

How is an "arch" identified in the OMPS data? How do we know when an arch shape might be geophysically accurate vs. one that is an artifact of a tiny cloud? Gteam describe a wholesale processing of the OMPS LP extinction profiles, applying the arch effect correction to the whole set. At least that is how I understood the method description. If this is the case, does that suggest that Gteam considers the arch effect a global

vulnerability? Much more clarification of the correction application is needed.

ANSWER 1: The first 7 pages of the Primary Concerns are based on a misinterpretation of Figure 3. We agree with the reviewer that the CCC was much larger than 180 km (this is evident, for example, from the Figures 12-14). But the arch effect occurs if the horizontal length of the cloud is less than 1100 km, and not 180 km (for a cloud height of 25 km). We improved Figure 3 and made additional Figure 3b, which clearly demonstrate the effect of the arch or the observed decrease in the height of the cloud when it is displaced from the tangential point. We have added the following text to the article:

"The arch effect is observed when the length of the visible part of the cloud is less than ~1100 km (at a cloud height of 25 km). Figure 3b shows cloud $F_0G_0$, 1 km thick and 226 km long, centered above tangential point T. Due to the curvature of the globe, such a cloud has an observed thickness of 2 km (see Figure 3b). If we take a cloud 226 km long and with a real thickness of N km, then the observed cloud thickness will be N+1 km. Thus, the real average height of a thin (1-2 km) cloud is underestimated by 0.5-1 km even under the most optimal observation conditions. Consider a cloud FG, the center of which is displaced from the tangential point by 273 km (or by $\varphi$~2.5 degrees). The real height of the FG cloud is 24-25 km, but its observed height varies from 13 to 22 km. If we consider the $F_0G$ cloud with a length of 499 km, then its real height above the earth's surface will be 24-25 km, and the observed height is 13-25 km. Let us take into account that the limb profiler assigns the latitude of the tangential point to any extended cloud. Therefore, a single cloud shown in Figure 3a in five different observed positions, instead of one real geographic latitude, receives several "observed" latitudes, which creates an arch effect. Let the region $F_0G_0$ be a gap in a continuous cloud. Then this gap, together with the arch effect, will lead to a decrease in the maximum observed height of the cloud layer by 1 km (see Figure 3b)."

I could find no discussion of the vulnerability of OMPS-LP extinction profiles to saturation in the presence of optically thick aerosol plumes. Given the likelihood that the Raikoke sulfates presented widespread scenes of such optically dense conditions, as did previous eruption plumes like Nabro, Sarychev Peak, and Kasatochi (Fomm et al., JGR, 2014; Lurton et al., https://doi.org/10.5194/acp-18-3223-2018) it is important for Gteam to directly address if/how this issue was dealt with in their various visualizations of OMPS LP extinction and SAOD.

ANSWER 2: Although Raikoke was the largest volcanic eruption seen by OMPS LP, it is still considered a mid-size eruption and nowhere near the size of Pinatubo or even El Chichon eruption. As noted by Rieger et al., (2019) (see below), the "saturation bias" cited by Fromm et al, 2014 and Lurton et al. 2018 was caused by the OSIRIS V5.0 algorithm conservative approach in masking any data when the extinction exceeded $2.5 \times 10^{-3}$ km$^{-1}$. OMPS LP algorithm has no such restriction, and we are not aware of any detector saturation caused by this volcanic plume.

Rieger, L. A., Zawada, D. J., Bourassa, A. E., and Degenstein, D. A.: A multi-wavelength retrieval approach for improved OSIRIS aerosol extinction retrievals, J. Geophys. Res.-Atmos., 124, 7286–7307, https://doi.org/10.1029/2018JD029897

Figure 14 and related discussion of CCC: Chouza et al (2020) https://doi.org/10.5194/acp-20-6821-2020 do a very similar tracking of the CCC (although they do not describe in CCC terms). It is not cited by Gteam. There appears to be a similarity in how the CCC is tracked but then a divergence occurs in late August. Because of the similarity and relevance to this paper, Gteam are encouraged to read Chouza et al. and evaluate the

similarities and discrepancies between the two treatments of the CCC.

ANSWER 3: We have added the following text to the article:

"Figure 15 was published in December 2019 at AGU-2019 (Gorkavyi et al, 2019). On September 24, 2019 the CCC was observed by lidars in Hawaii (Chouza et al, 2020). Chouza et al (2020) traced the trajectory of this cloud back to July 17, 2019. Although the two studies were done independently, they came up with very similar results. Chouza et al (2020) consider this cloud as a Raikoke plume, but we prefer to call it CCC because it is a very small part of the Raikoke plume."

Kloss et al. (2021) present a SAOD analysis very similar to Figure 7. In fact, the same version of OMPS LP retrieval is used for both. I did not see in GTeam what I expected, an acknowledgement of this previously published analysis and a motivation for presenting the analysis anew. The only substantial differences in the Gteam plot are an addition of two markers for pyroCb sources and a slightly later endpoint. Neither of these visual differences are taken advantage of, from my reading. If I missed it, please advise. Regardless, it would be important for Gteam to assess whether the figure should stay and also cite Kloss et al. for the earlier rendition.

ANSWER 4: We have replaced Figure 7 with plots of OMPS LP aerosol extinction at three different altitudes, 14.5, 18.5, and 20.5 km.

**Minor Concerns**

Note: Text below in quotation marks is directly from Gteam.

P4, L118: The schematic (Figure 3) shows a "feature" that is limited both horizontally and vertically. Why does this statement call out only the vertically limited aspect? What's the relative importance of limitations in the vertical versus horizontal?

ANSWER 5: See Answer 1

P4, L122: "If we believe that these lower altitude values do not represent a true aerosol signal…" How is this belief ascertained?

ANSWER 6: If we compare the lidar observations of the CCC (Figure 14) and the limb observations of this cloud (Figure 4), it becomes clear that all parts of the arch below 23 km are artifacts. There are many such examples. For example, PMCs are always 80 to 85 km high. According to the data of the limb sensor (see below, unpublished), these clouds have apparent heights of up to 25 km. All parts of the PMS observed below 80 km are artifacts from the arch effect.

[Figure]

Figure 3 caption: The caption needs to describe the meaning of the horizontal solid and dash-dot line. Presumably these show

the OMPS instrument vertical FOV. But if so, why are the line styles different?

ANSWER 7: Now all lines are similar.

Figure 4. Please add an x-axis marking/labeling in deg. latitude, to give the reader a useful geographic refence frame.

ANSWER 8: Done

P6, L153: The Junge layer peaks much higher according to Kremser https://doi.org/10.1002/2015RG000511 citing Junge.  An examination of CALIOP data shows lots of Raikoke aerosols widespread, lower down, on both sides of the CCC.

ANSWER 9: Our statement "The aerosol layer at an altitude of 18-20 km with a decrease to the north to 15-16 km is the Junge layer" (P6, L153)" is in full accordance with LARC/NASA data (see figure below) and does not contradict the classical works: "stratospheric aerosol occurs in a distinct layer between 15 and 25 km altitude with a peak near 20 km [Junge et al., 1961]" (Kremser et al, https://doi.org/10.1002/2015RG000511). Also we added in text next statement after Fig.4:
"Note that CCC has been the southernmost part of the Raikoke plume since late July 2019 (see section 3.3). During this period, only the usual Junge aerosol layer was located south of latitude 20°N."

[Figure]

P6, L158: Please define/describe "artifact density." Why is the term "density" used? Why not "extinction?"

ANSWER 10: "Density" was replaced by "extinction".

P6, L157-161: This paragraph has several vague or unintuitive terms, such as "real," "artifact density," "cleaned." Please replace these terms or define them.

ANSWER 11: "real" was replaced by "true"; "density" -> "extinction"; "cleaned" -> "corrected".

P6, L158: How is an "isolated" feature determined from OMPS data alone? There is no discussion of complementary data (e.g. OMPS nadir) informing the limb data.

ANSWER 12: We use CALIOP (and SAGE – see new Fig. 10) as an additional source of information.

P7, L185: "The sensitivity of the satellite data we examined…" What satellite data? OMPS NM? According to Figures 5 and 7, the aerosols are detected much longer.

ANSWER 13: We improve our statement: "The sensitivity of the satellite data (OMPS LP, CALIOP) we examined is such that the CCC from the Raikoke volcano was observable for 3 months following the eruption and the increase in the Junge aerosol layer was observed even longer."

Figure 5 caption: "5: The daily zonal mean (45-85N) SO2 mass (assuming a cloud height of 13 km) and the aerosol extinction coefficient at 675 nm (13-18 km.)." This is 6 OMPS LP altitude bins. Which extinction is it? Max, min, average?

ANSWER 14: We have clarified the caption for Fig. 5: "the average aerosol extinction coefficient at 675 nm (summed up over 13-18 km and divided by 6 (km) to get the average extinction coefficient)."

Figure 5 caption: "ash fallout" How does the reader know how much ash fallout is observed? What does ash fallout have to do with SO2? Please explain.
ANSWER 15: The statement about "Ash fallout" was deleted.

Figure 5 caption: "…extinction starts to decrease due to gravitational sedimentation, but very slowly (from OMPS LP and NM data)" What does this mean? How do these inform about sedimentation? Sedimentation is one of several processes that an diminish extinction. How is sedimentation demonstrated herein?
ANSWER 16: We have clarified: "decrease due to gravitational sedimentation and other processes"

Figure 6 caption: "main plume" How is the "main plume" defined?" This is the first invocation of "main plume."

ANSWER 17: We have clarified: "the main plume (45-85°N)"

Figure 6 caption: "south branch" Is "south branch" synonymous with CCC, or is it more general? Perhaps just stick with "CCC" if they are one and the same.

ANSWER 18: "south branch" was replaced by CCC.

P9, L227: "After that, the detection limit of the instrument…" What id the detection limit? Please state the value.

ANSWER 19: We added next statement: "OMPS in the stratosphere can typically detect 0.2-0.3 DU of SO2".

P9, L239: No pyroCbs are mentioned by Kloss et al. as far as I can tell. If they are, please provide some more detail. If not, what is the justification for adding these symbols to the figure? There is no evident increase of SAOD clearly attributable to these symbols in the plot.

ANSWER 20: We are fixed the Fig. 7 with a caption and have clarified:

"In addition, Figures 7b and c also show the aerosol transport to subtropics and tropics at higher altitudes. Increased aerosol loading in the lower stratosphere can also be attributed to two pyroCumulonimbus (pyroCb) events that took place before and after the eruption, Alberta fires (June 18) and Siberian fires (July 2) (Kloss et al., 2021). OMPS LP detected both plumes in the stratosphere at 12-13 km, although it became difficult to separate them from Raikoke plume once it spread around the NH."

P10, L241: "The top panel in Fig. 8 shows that the maximum altitudes of the plume are around 25 km, when the plume

penetrates the tropics." How is "when" shown? Figure 7 seems to show that Raikoke material only got to the subtropics.

ANSWER 21: We have replaced Figure 7 with plots of OMPS LP aerosol extinction at three different altitudes, 14.5, 18.5, and 20.5 km. The new figure clearly shows the plume transport to the tropics at altitudes 18.5 and 20.5 km.

Figure 8(a): Please explain the two apparent outliers of ~25 km layers over Northern Europe. They seem to be all by themselves; no indication of a ramp up to those altitudes from nearby layer observations.

ANSWER 22: We agree with the reviewer that the original figure was confusing. We have now modified the figure to show the day number instead of the maximum altitude since the reader can match the altitude in Fig8b with the location and day number in Fig 8a. In addition, we changed the maximum altitude period to only show the first four months of the plume altitude, which is 150 days since June. The maximum altitude estimate uncertainties increases as the plume move around and subside following the first 150 days. Fig 8a period now matches Fig8b white line, which wasn't the case in the previous version. The only difference is that Fig8b shows the location of the plume every two days for 150 days, while Fig8b is plotted every day for the same period. We have now modified the text accordingly.

Figure 8(a) caption: What is the number of days shown? Span of time? Are all the reported dates consecutive, or are there gaps? If there are significant gaps, these dates should be mentioned.

ANSWER 23: See our reply to the previous comment. The new caption and figure are clearly showing the period, which is 150 days since June, plotted every other day.

Figure 8 caption: The sAOD panel is not described.

ANSWER 24: We have added the following to the figure's caption "Figure 8(b): (top) Stratospheric aerosol optical depth (x $10^3$, sAOD) at 675 nm for latitudes and period similar to Figure 8b. The sAOD is derived by integrating aerosol extinction profiles above the tropopause to 35 km"

Figure 8(b) caption: How are the data comprising the white line determined? The white line doesn't conform closely to the color shading. I.e. it crosses extinction contours in time. Please explain fully.

ANSWER 25: The plume altitude is derived using the OMPS LP cloud algorithm, which can identify enhanced aerosol layers in the stratosphere. We have now added this to the text. The reason for the mismatch between the white line and the extinction contour lines is the color table used, which favors larger aerosol extinction values. We have now changed the color table to better show small extinction values.

Figure 9 caption: The Junge layer is higher than the range indicated here.

ANSWER 26: see Answer 9.

P16, L309: Specifically, how were the CCC detections done out to 22 September? I could not find a definitive explanation.

ANSWER 27: We added after Fig.13 (new): "Using the data of CALIOP, OMPS LP and NM as well as OMI"

Figure 14 caption: "OMI" is mentioned here for the first time as part of this analysis, but it is not mentioned in the Data and Methods section. This should be done, unless OMI is not part of the present analysis.

ANSWER 28: See ANSWER 27, also we added to end 2.1: "We also used OMI data as an additional source."

P18, L349: "A carbonaceous aerosol plume associated with wild fires in British Columbia in August 2017 reached the stratosphere a few days following the initial injection above the tropopause…" Confusing. It is stated that the plume was injected above the tropopause, but only reached the stratosphere a few days later. Both clauses are attributed to Torres et al. Please clarify.

ANSWER 29: The text has been changed: "A carbonaceous aerosol plume associated with wild fires in British Columbia in August 2017 reached the stratosphere a few days following the initial injection and resulting from self-lofting triggered by solar heating (Torres et al., 2020; see similar effect for Australian fires - Khaykin et al., 2020)."

P18, L352: "…19 to 26 kilometers is observed for a cloud consisting of sulfate aerosol." From the material presented to this point, how does the reader know it is sulfate? If compositional information has not been provided, a citation is needed. Alternatively, the paper could just state that sulfate is assumed.

ANSWER 30: The text has been changed: "observed for a cloud presumably consisting of sulfate aerosol"

P18, L376: What are "large-scale mixing events?" Please explain using a meteorological perspective.

ANSWER 31: The text has been changed: "large-scale mixing events" -> "turbulent mixing"

P18, L279: "SO2 reduction" is vague and general. Provide more detail to this description, such as "e-folding time"

ANSWER 32: The text has been changed: "SO2 reduction" -> "e-folding time"

P18, L383: "…the CCC circled the globe almost three times at the latitude of 30oN…" The CCC ended up almost 10 deg. south of this according to figures herein. Perhaps elaborate a little more on the CCC's slow movement even south of 30.

ANSWER 33: We added: "(during September, the CCC shifted south to 15-20$^{\circ}$N)"

P18, L383: "…increased its height from 19 to 28 km." "28" is higher than shown if Fig. 14.  The text in the body of the paper only gives pressure and theta. Suggestion: enhance the description of the endpoint altitude in the main text and state "26 km" or whatever actual number the authors think is represented by the figure.

ANSWER 34: The text has been changed: "28 km" -> "26 km"

P18, L387-389: This sentence is confusing, combining 3 ideas: arch effect, 45 km background, and aerosol that can pollute the background. Please flesh out these points more exhaustively.

ANSWER 35: We transfer this sentence to end 2.2 with clarification.

More generally, the Conclusion section is too thin, and doesn't flow logically from points made in the body. A thorough rework of this section is called for.

ANSWER 36: The conclusion has been improved and expanded.

ANSWER 37: We have corrected all errors and took into account all the comments of the reviewer made in the comments to the text of the article.

---

## Referee Report (RR1)

**Review of amt-2021-58**

The authors have addressed most of my initial concerns, however I still take issue with the implemented "arch effect" correction which I believe needs to be considered before the manuscript can be published.

**General Comments**

In my opinion the correction is not really necessary for the analysis that follows, but if it is to be included then statements made about it need to be justified which in many places they are not. To be clear, the authors use the correction to attempt to improve the resulting extinction profiles in two, somewhat coupled, ways: first to get a better sense of the locality of the Raikoke plume, and second to get a better extinction estimates from OMPS-LP. The first point is fine, I take no issue with, and might be important for the discussion for the CCC. The second point is the one that I have a problem with. There is no evidence presented that the correction is improving the extinction, and in fact I believe it is making it worse. As I stated previously my intuition is that a 1D retrieval is smoothing the true field into the shape of an arch, not simply introducing unphysical arches. This means that the overall extinction loading is relatively unaffected by the biases in a 1D retrieval, and yet the correction is significantly reducing the loading. Now, I fully admit that I might be wrong, but there is no justification at all in the manuscript for the efficacy of the correction and there are figures in the manuscript that suggest the correction is not working (see specific comments).

If justification for the correction cannot be provided then my preference would be to remove it, or at least use it only to assess the locality of the plume and not the overall loading. As I previously stated, I would also be fine if the correction was presented as a potential source of error instead of an actual correction. This is still okay, but it would mean that the correction is not applied and the differences are only used as uncertainty estimates. The authors additional statement that "We consider this procedure of correction only as an estimate" does not go far enough since they then continue to use the corrected extinction as gospel.

I have a few specific comments below, all related to the arch effect.

**Specific Comments**

**p.5 l.153:** The statement "If we believe that these lower altitude values do not represent a true aerosol signal, we need to apply a correction in order to accurately determine overall aerosol loading." neglects the fact that a 1D retrieval is probably smoothing the true field, reducing the extinction inside the plume in addition to creating arches. Peforming the correction will do more harm than good if the 1d effect is more of a smoothing effect.

**p.8 l.210:** The statement "The arch effect characteristic of the limb observations should be taken into account when calculating the optical thickness of aerosol clouds" simply is not justified with any of the information presented in the manuscript. If the effect is to be included there needs to be some evidence provided that it is improving the result.

**Figure 5:** I have the same comment I had last time since I do not believe it was sufficiently answered. Why is the correction reducing the extinction by ∼50% before the eruption even happens? To me this is evidence that the correction is not working as it should. Essentially this is saying that every limb sounder is overestimating aerosol extinction by upwards of 30-50% in clear conditions?

**Figure 10:** This is maybe more of a comment. SAGE III is affected by the same 1D retrieval bias, the authors are suggesting that it could be biased by 30-40%, so why is it usable here? I realize the same correction cannot be applied to the sparse sampling of SAGE III, but the fact that the manuscript does not mention this issue seems odd if it is as important as the authors claim.

**p.22 l.469:** "We have shown that this effect is significant" the only way to show this effect is significant is to simulate 2D radiances and then retrieve in 1D. You have shown that a very specific correction technique, which may or may not work, introduces significant differences.

---

## Referee Report (RR2)

Review of GTeam response to reviewers and new manuscript. By Mike Fromm

My summary assessment is that this manuscript merits publication in AMT if GTeam strengthen their "arch effect" application to the very complex lower stratospheric, freshly perturbed volcanic aerosol condition that is the focus of this paper, in accord with suggestions detailed below.

GTeam have made substantial improvements and responded to every reviewer comment. The revised manuscript is greatly improved. In particular, GTeam have bolstered and clarified their treatment of the "arch effect," which is the paper's primary new contribution to the AMT literature in my assessment. Particularly helpful to the reviewer was their illustration of the arch effect for a mesospheric cloud. GTeam succeeded in clarifying how lower stratospheric clouds might produce the arch effect.

However, in doing so, GTeam's response to reviewers and the revised manuscript elucidate the reviewers' original concerns. By using the PMC as an example, GTeam nicely show how a simple cloud of limited vertical extent, horizontally flat, and singular in the view of OMPS-LP may be well represented by the model they construct in Figure 3. But this example is vastly different from the preponderance if scenes in the lower stratosphere post Raikoke. Based strictly on the CALIOP curtains shown by GTeam in Figures 12 and 14, it is obvious that multiple, stacked sulfate layers were the rule in the post-Raikoke stratosphere, unlike tie simple, single-layer geometry in the case of PMCs.  That's not to say that the arch effect doesn't occur for lower stratospheric clouds, it is just to say that a convoluted, multi-layer condition (such as Fig's 12 and 14) either compromises the arch-effect interpretation or could even create a false arch effect if the cloud is itself arched or sloped in any manner. An example is the Raikoke sulfate layer sampled by CALIOP over northern Canada, seen here.

 https://www-calipso.larc.nasa.gov/products/lidar/browse_images/show_v4_detail.php?s=production&v=V4-

10&browse_date=2019-07-06&orbit_time=10-50-31&page=4&granule_name=CAL_LID_L1-Standard-V4-10.2019-07-06T10-50-31ZD.hdf

This goes to my original question as to statements such as this on line 154: "If we believe that these lower altitude values do not represent a true aerosol signal…" There is no robust basis for relying on said belief when it is straightforward to test such suppositions with complementary (and regularly near coincident) data like CALIOP.  Specific to the presumed OMPS-LP arch effect displayed in Fig. 4, CALIOP data in Fig. 14 make clear that indeed lower volcanic layers were north, south, and underneath the CCC that GTeam suggest was responsible for the arch effect in Fig. 4.

Given the clarification GTeam provide in Fig. 3 and the recognition that the example used herein (the 31 August CCC) for the arch effect is far from ideal (noting the large extinction of the CCC feature and the additional layers below), my suggestion is for GTeam to select another scene with which to make the arch-effect case. The ideal scene would be akin to the PMC example they trust: flat in real altitude and singular from the OMPS viewpoint.

CALIOP would provide the independent standard, with an unambiguously single-layer, non-sloping cloud of horizontal extent consistent with the argument and model manifested in Fig. 3.

Regarding my question in the original review of saturation effect on OMPS extinction, I recognize that the OSIRIS situation is technically different from OMPS, but it is still apparent that OMPS (and all other historical limb-view sensors) cannot faithfully characterize extinctions across full range in the landscape of the Raikoke plume. The 31cAugust, 1 September 2019 CCC case was mentioned in the first review. According to CALIOP backscatter, the actual maximum extinction presented to OMPS in scenes such as this far exceeds the maximum OMPS extinction in the OMPS record. The OMPS data I have at hand (about a year's worth of global profiles) doesn't represent the full OMPS record, but it appears that extinctions do not exceed values of about .005/km at 675 nm.  Given that CALIOP-based evidence speaks to much larger values being commonplace in the Raikoke plume, some discussion of OMPS low bias, for whatever the correct technical reason, should be presented.

Regarding my concerns with the Junge layer attribution in parts of the manuscript, I remain unconvinced by GTeam's explanation. By definition, the Junge layer is a background condition. The features that GTeam attribute to the Junge layer are definitely enhancement to the background (The figure they present in the author responses is quantifiably vague.). GTeam claim that there were no volcanic aerosols south of about 20N, but clearly there were, likely attributable to Ulawun, which is clearly shown in the paper. Hence in the period analyzed herein, there were simultaneous stratospheric sulfate aerosol layers from northern tropics to high latitudes.

Upon this re-review, I noticed that GTeam stated that due to instrument sensitivities, the Raikoke aerosols were only detected for 3 months. This is incompatible with several figures shown herein. If this assessment is to remain, it must be attributed to specific visualizations herein and not contradicted by any other.

---

## Author Response (AR2)

**NEW ANSWERS to RC1** (07/30/2021)

Review 1 of amt-2021-58

The authors have addressed most of my initial concerns, however I still take issue with the implemented "arch effect" correction which I believe needs to be considered before the manuscript can be published.

General Comments

In my opinion the correction is not really necessary for the analysis that follows, but if it is to be included then statements made about it need to be justified which in many places they are not. To be clear, the authors use the correction to attempt to improve the resulting extinction profiles in two, somewhat coupled, ways: first to get a better sense of the locality of the Raikoke plume, and second to get a better extinction estimates from OMPS-LP. The first point is fine, I take no issue with, and might be important for the discussion for the CCC. The second point is the one that I have a problem with. There is no evidence presented that the correction is improving the extinction, and in fact I believe it is making it worse. As I stated previously my intuition is that a 1D retrieval is smoothing the true field into the shape of an arch, not simply introducing unphysical arches. This means that the overall extinction loading is relatively unaffected by the biases in a 1D retrieval, and yet the correction is significantly reducing the loading. Now, I fully admit that I might be wrong, but there is no justification at all in the manuscript for the efficacy of the correction and there are figures in the manuscript that suggest the correction is not working (see specific comments). If justification for the correction cannot be provided then my preference would be to remove it, or at least use it only to assess the locality of the plume and not the overall loading.

ANSWER 1:  We have added to the new version of the article a new figure 4b and its discussion:

[Figure]

[Figure]

Fig 4b. (Left) An arch that appears during the limb observation of an aerosol cloud 2 km thick (H = 23-25 km) and 2° long. Each frame corresponds to a satellite orbital shift by 1°. Radiance from aerosol is proportional to the number of observed particles, taking into account the distance to them (the left part of the arch is slightly brighter than the right, because on frames 17-21 the cloud was closer to the satellite than on frames 23-27). The radiance units are arbitrary. (Right). The red line is the profile of a cloud of 400 particles, which is observed on frame 22 (middle of the arch in Fig. 4a). The black line is the number of visible particles, summed over all frames, that is, over the entire arch.

"The reality of the discussed "arch effect" is confirmed by simple modeling. Arch model in Fig. 4b was obtained by direct modeling of a cloud of 400 particles located at the nodes of a uniform grid (20 particles are distributed at a length of 2 degrees, and 20 rows of particles are uniformly distributed along the radius in the range of 23-25 km). The model does not use radiative transfer models. Fig. 4b (left) only the distance between the particles and the satellite is taken into account, while Fig. 4b (right) shows the simply visible (for the limb sensor) number of particles at a given altitude h (in 1 km step). The red line corresponds to a one-time observation of a cloud located at the tangent point at an altitude of h = 22-25 km (an increase in the apparent thickness of the cloud by 1 km is associated with the curvature of the Earth, which is why the cloud itself turns out to be curved - see Fig. 3). That is, it is the most realistic observation of the cloud at its optimal location. The black line shows the sum of the cloud particles observed at different times. As a result of this summation, the number of particles visible at a given observed height h (which differs from the real constant cloud height H = 23-25 km) turns out to be overestimated. Therefore, when analyzing the picture in Fig. 4, we must remember that it is composed of frames received at different times, so the one cloud will be registered many times. The same effect of multiple registration will be observed for clouds of any complexity and configuration, including a uniform aerosol layer, because any cloud can be divided into a large number of elementary pieces, similar to a simple compact cloud considered in Fig. 4b.

The considered model of the "arch effect" does not depend on the specific model of radiation transfer and the methods of retrieval of the spatial distribution of aerosol. Therefore, for specific limb sensors (OMPS-LP, SAGE III, OSIRIS), it is necessary to evaluate how accurately the available retrieval packages handle compact clouds. This is especially true for 1D retrieval methods, which assume spherical symmetry of the atmosphere and which are used to obtain aerosol extinction in the OMPS/LP. Obviously, the arch effect for a spherically uniform aerosol layer should be fully compensated by the 1D retrieval model. But the further the real system is from the spherical symmetry, the more difficult it will be to take into account the "arch effect"".

Discussion of the problems of various retrieval and RTM for limb sensors is beyond the scope of this article. We have shown the importance of the "arch effect" on a simple model and in one specific case (as a maximal estimation) and raised the question of how effectively the various retrieval methods deal with this problem. If we exclude the discussion and assessment of the "arch effect", then the problem of the efficiency of processing compact clouds is out of the field of view of researchers.

As I previously stated, I would also be fine if the correction was presented as a potential source of error instead of an actual correction. This is still okay, but it would mean that the correction is not applied and the differences are only used as uncertainty estimates. The authors additional

statement that "We consider this procedure of correction only as an estimate" does not go far enough since they then continue to use the corrected extinction as gospel. I have a few specific comments below, all related to the arch effect.

ANSWER 2: We are grateful to the referee for this important discussion, which helped to clarify a lot about the "arch effect". We hope that the new edition of the article (see also Answers 1 and 2) has taken into account all the main comments of the reviewer.

**Specific Comments**

p.5 l.153: The statement "If we believe that these lower altitude values do not represent a true aerosol signal, we need to apply a correction in order to accurately determine overall aerosol loading." neglects the fact that a 1D retrieval is probably smoothing the true field, reducing the extinction inside the plume in addition to creating arches. Performing the correction will do more harm than good if the 1d effect is more of a smoothing effect.

ANSWER 3:  In the new version of the article, we have separated the "arch effect" from the possible problems of retrieval methods. The effectiveness of specific techniques in relation to the "arch effect" should be examined on a case-by-case basis (for different limb sensors and retrieval and RTM package).

p.8 l.210: The statement "The arch effect characteristic of the limb observations should be taken into account when calculating the optical thickness of aerosol clouds" simply is not justified with any of the information presented in the manuscript. If the effect is to be included there needs to be some evidence provided that it is improving the result.

ANSWER 4:  See Answer 1 and new addition in the paper.

Figure 5: I have the same comment I had last time since I do not believe it was sufficiently answered. Why is the correction reducing the extinction by ~50% before the eruption even happens? To me this is evidence that the correction is not working as it should. Essentially this is saying that every limb sounder is overestimating aerosol extinction by upwards of 30-50% in clear conditions?

ANSWER 5:  We update Fig.5 adding arrows as error bars. Also next statement added in the new version of the paper (see also Answer 1).:

"To estimate the possible retrieval uncertainty due to the "arch effect", we apply a simple compensation method for one specific case of the Raikoke aerosol cloud (see next section). This compensation method assumes that the considered aerosol clouds form compact clusters or a highly heterogeneous aerosol layer, and that the arch effect was not taken into account in retrieval. Thus, this example should be regarded as a maximal estimation for the "arch effect". Where this assumption is not valid, our correction will be overestimated, as, for example, happened with the correction of the background aerosol value (see Fig. 5), which was observed before the Raikoke eruption".

Figure 10: This is maybe more of a comment. SAGE III is affected by the same 1D retrieval bias, the authors are suggesting that it could be biased by 30-40%, so why is it usable here? I realize

the same correction cannot be applied to the sparse sampling of SAGE III, but the fact that the manuscript does not mention this issue seems odd if it is as important as the authors claim.

ANSWER 6: We not rejected any OMPS-LP and SAGE III aerosol limb data: we used this data in seven figures in our paper. But we discuss "arch effect" and apply this effect for estimation maximal possible errors for one specific case (and for one graph on Fig 5). This estimation is show the problem which must be investigated in future (see Answer 2).

1p.22 l.469: "We have shown that this effect is significant" the only way to show this effect is significant is to simulate 2D radiances and then retrieve in 1D. You have shown that a very specific correction technique, which may or may not work, introduces significant differences.

ANSWER 7: See Answer 1 and new addition in the paper

**NEW ANSWERS to RC2**

**Review of GTeam response to reviewers and new manuscript. By Mike Fromm**

My summary assessment is that this manuscript merits publication in AMT if GTeam strengthen their "arch effect" application to the very complex lower stratospheric, freshly perturbed volcanic aerosol condition that is the focus of this paper, in accord with suggestions detailed below.

GTeam have made substantial improvements and responded to every reviewer comment. The revised manuscript is greatly improved. In particular, GTeam have bolstered and clarified their treatment of the "arch effect," which is the paper's primary new contribution to the AMT literature in my assessment. Particularly helpful to the reviewer was their illustration of the arch effect for a mesospheric cloud. GTeam succeeded in clarifying how lower stratospheric clouds might produce the arch effect.

However, in doing so, GTeam's response to reviewers and the revised manuscript elucidate the reviewers' original concerns. By using the PMC as an example, GTeam nicely show how a simple cloud of limited vertical extent, horizontally flat, and singular in the view of OMPS-LP may be well represented by the model they construct in Figure 3. But this example is vastly different from the preponderance if scenes in the lower stratosphere post Raikoke.

ANSWER 8:  We have added to the new version of the article a new figure 4b and its detailed discussion (see ANSWER 1). In the new version of the article, we examined a cloud in the stratosphere at an altitude of 23-25 km, corresponding to the Raikoke case (see Fig. 4b).

Based strictly on the CALIOP curtains shown by GTeam in Figures 12 and 14, it is obvious that multiple, stacked sulfate layers were the rule in the post-Raikoke stratosphere, unlike tie simple, single-layer geometry in the case of PMCs.  That's not to say that the arch effect doesn't occur for lower stratospheric clouds, it is just to say that a convoluted, multi-layer condition (such as Fig's 12 and 14) either compromises the arch-effect interpretation or could even create a false arch effect if the cloud is itself arched or sloped in any manner. An example is the Raikoke sulfate layer sampled by CALIOP over northern Canada, seen here.
https://wwwcalipso.larc.nasa.gov/products/lidar/browse_images/show_v4_detail.php?s=production&v=V4-10&browse_date=2019-07-06&orbit_time=10-50-31&page=4&granule_name=CAL_LID_L1-Standard-V4-10.2019-07-06T10-50-31ZD.hdf

This goes to my original question as to statements such as this on line 154: "If we believe that these lower altitude values do not represent a true aerosol signal…" There is no robust basis for relying on said belief when it is straightforward to test such suppositions with complementary (and regularly near coincident) data like CALIOP.  Specific to the presumed OMPS-LP arch effect displayed in Fig. 4, CALIOP data in Fig. 14 make clear that indeed lower volcanic layers were north, south, and underneath the CCC that GTeam suggest was responsible for the arch effect in Fig. 4.

Given the clarification GTeam provide in Fig. 3 and the recognition that the example used herein (the 31 August CCC) for the arch effect is far from ideal (noting the large extinction of the CCC feature and the additional layers below), my suggestion is for GTeam to select another scene with which to make the arch-effect case. The ideal scene would be akin to the PMC example they trust: flat in real altitude and singular from the OMPS viewpoint. CALIOP would provide the independent standard, with an unambiguously single-layer, non-sloping cloud of horizontal extent consistent with the argument and model manifested in Fig. 3.

ANSWER 9: Discussion in new version of the paper covers these topics. The same "arch effect" will be observed for clouds of any complexity and configuration, including a uniform aerosol layer, because any cloud can be divided into a large number of elementary pieces, similar to a simple compact cloud considered in Fig. 4b. We plan to analyzed different cases from these reviewer's comments in next paper (preliminary we did this for simple case: see below that for long cloud ~10° we have not "arch" but "mushroom", but decide not overloaded current paper).

[Figure]

Fig. Arch effect for long ~10° cloud (some conditions as on Fig.4b)

Regarding my question in the original review of saturation effect on OMPS extinction, I recognize that the OSIRIS situation is technically different from OMPS,

but it is still apparent that OMPS (and all other historical limb-view sensors) cannot faithfully characterize extinctions across full range in the landscape of the Raikoke plume. The 31cAugust, 1 September 2019 CCC case was mentioned in the first review. According to CALIOP backscatter, the actual maximum extinction presented to OMPS in scenes such as this far exceeds the maximum OMPS extinction in the OMPS record. The OMPS data I have at hand (about a year's worth of global profiles) doesn't represent the full OMPS record, but it appears that extinctions do not exceed values of about .005/km at 675 nm.  Given that CALIOP-based evidence speaks to much larger values being commonplace in the Raikoke plume, some discussion of OMPS low bias, for whatever the correct technical reason, should be presented.

ANSWER 10:  While there is no hard cutoff limit in the OMPS LP retrieval algorithm, there is a restriction on how much the retrieved aerosol extinction is allowed to grow per iteration at each altitude relative to the first guess, (see Loughman et al., 2018); Taha et al., 2021), which "may" cause an underestimation of the retrieved aerosol if the aerosol is too large. This constrain was further relaxed in the V2.0 algorithm (Taha et al., 2020). The CCC case was an interesting one given that the plume was a mixture of $SO_2$ and aerosol (as confirmed by TROPOMI and OMPS NM), which raises questions about the ability of limb scattering instruments to accurately measure $SO_2$. In any case, more detailed work is needed to validate the algorithm performance during large eruptions and pyroCbs, which is subject to a separate study and beyond the scope of this work.

The following sentence is now added after Fig.10 "Given that this plume is mostly composed of $SO_2$ and aerosol (Fig 12), it is likely that OMPS LP is underestimating its magnitude"

Regarding my concerns with the Junge layer attribution in parts of the manuscript, I remain unconvinced by GTeam's explanation. By definition, the Junge layer is a background condition. The features that GTeam attribute to the Junge layer are definitely enhancement to the background (The figure they present in the author responses is quantifiably vague.).

ANSWER 11: We have changed the term "Junge layer" to more general "aerosol layer" or "background aerosol layer".

GTeam claim that there were no volcanic aerosols south of about 20N, but clearly there were, likely attributable to Ulawun, which is clearly shown in the paper. Hence in the period analyzed herein, there were simultaneous stratospheric sulfate aerosol layers from northern tropics to high latitudes.

ANSWER 12: We deleted the statement "During this period, only the usual Junge aerosol layer was located south of latitude 20°N".

Upon this re-review, I noticed that GTeam stated that due to instrument sensitivities, the Raikoke aerosols were only detected for 3 months. This is incompatible with several figures shown herein. If this assessment is to remain, it must be attributed to specific visualizations herein and not contradicted by any other.

ANSWER 13: This statement was corrected: "The sensitivity of the satellite data (OMPS LP, CALIOP) we examined is such that the aerosol cloud from the Raikoke volcano was observable for many months following the eruption."

---

## Author Response (AR3)

**M. Fromm**

GTeam have generally responded well to the latest round of reviews and improved the manuscript.

I will defer to Reviewer 1 for how well GTeam addressed his/her concerns regarding the :arch effect" and the modified wording of the extinction correction. The changes seemed adequate to me.

There is one answer GTeam gave (Answer 10) that I have a concern with. I had made the point that the OMPS-LP extinctions may have a low bias and asked them to consider discussing that. In their Answer 10 (and the textual revision made) GTeam presented two perplexing points. One is that they refer to Taha et a; (2021) and version 2.0 of the OMPS-LP aerosol version. But that is irrelevant for this paper because they use v1.5, which they acknowledge may have extinction underestimates for the densest Raikoke plume elements. Taha et al. (2021) is not cited nor discussed in this manuscript. The second perplexing part of their answer is that the CCC in question is an interesting case because of its composition being a blend of gaseous SO2 and sulfate particles. But this is not peculiar to the CCC; much of the plume was a combination of SO2 and sulfates. Moreover, OMPS-LP scattering is only sensitive to particles. So introducing the SO2 element in their response and the manuscript, in the discussion of extinction bias, has no apparent merit. Hence I would suggest they revise their newly added sentence to remove the reference to SO2, and limit it to a statement about the possible low bias in v1.5 extinctions where the plume is particularly concentrated.

Authors:

We deleted on L374 the sentence "Given that this plume is composed of a mixture of SO2 and aerosol (Fig 12), it is likely that OMPS LP is underestimating its magnitude." And added on L378 the following "The OMPS LP version 1.5 algorithm has a restriction on how much the retrieved aerosol extinction is allowed to grow per iteration at each altitude relative to the first guess, which may cause an underestimation of the retrieved aerosol where the plume is concentrated (README Document for the Suomi-NPP OMPS LP L2 AER675 Daily Product, 2019)." We also added the README reference to the reference list.

**Reviewer 1**

The authors have done a good job in taking into my previous suggestions. I believe the manuscript is greatly improved and can be published subject to one technical correction and one comment that the authors can choose to take into account if they wish:

Comment:

l.219: "...including a uniform aerosol layer..." I was very confused when I first read this but after reading the next paragraph it made sense and perhaps illuminated the reasons for some of the initial confusion around the arch effect between myself and the authors. The arch effect as described in this paragraph is an effect on the observed radiances, and not the 1D retrieval as I had originally assumed. One way to remove the arch effect is a full 2D retrieval, but a 1D retrieval also removes the arch effect for a horizontally homogenous atmosphere which explains the difference in background conditions. I will leave it up to the authors to decide if any additional explanation should be added here.

Authors: We have moved from L226 statement

"Obviously, the arch effect for a spherically uniform aerosol layer should be fully compensated by the 1D retrieval model. But the further the real system is from the spherical symmetry, the more difficult it will be to take into account the "arch effect""

to L219 - right after the phrase about "a uniform aerosol layer", which makes the logic clearer.

Technical Correction:

l.289: arc -> "arch"

Authors: Fixed.